# Diffusion-limited reactions in dynamic heterogeneous media

Yann Lanoiselée[1], Nicolas Moutal[1] & Denis S. Grebenkov [1]

Most biochemical reactions in living cells rely on diffusive search for target molecules or regions in a heterogeneous overcrowded cytoplasmic medium. Rapid rearrangements of the medium constantly change the effective diffusivity felt locally by a diffusing particle and thus impact the distribution of the first-passage time to a reaction event. Here, we investigate the effect of these dynamic spatiotemporal heterogeneities onto diffusion-limited reactions. We describe a general mathematical framework to translate many results for ordinary homogeneous Brownian motion to heterogeneous diffusion. In particular, we derive the probability density of the first-passage time to a reaction event and show how the dynamic disorder broadens the distribution and increases the likelihood of both short and long trajectories to reactive targets. While the disorder slows down reaction kinetics on average, its dynamic character is beneficial for a faster search and realization of an individual reaction event triggered by a single molecule.

[1] Laboratoire de Physique de la Matière Condensée (UMR 7643), CNRS—Ecole Polytechnique, University Paris-Saclay, 91128 Palaiseau, France. Correspondence and requests for materials should be addressed to D.S.G. (email: denis.grebenkov@polytechnique.edu)

Diffusion is the central transport mechanism in living cells and, more generally, in biological systems. Molecular overcrowding, cytoskeleton polymer networks, and other structural complexities of the intracellular medium lead to various anomalous features such as nonlinear scaling of the mean square displacement (MSD), weak ergodicity breaking, non-Gaussian distribution of increments, or divergent mean first-passage times (FPT) to reactive targets[1–10]. These features are often captured in theoretical models via long-range correlations (e.g., fractional Brownian motion or generalized Langevin equation), long-time caging (continuous time random walks), or hierarchical structure (diffusion on fractals)[11–18]. While the impact of heterogeneity of the medium[19–21], and of reactive sites[22,23] onto diffusion and the macroscopic reaction rate was investigated, the diffusivity of a particle was usually considered as constant. However, the structural organization of living cells and other complex systems such as colloids, actin gels, granular materials, and porous media suggests that the diffusivity can vary both in space and time.

Several recent studies were devoted to such heterogeneous diffusion models. At the macroscopic level, the dynamics and the reaction kinetics can still be described by the Fokker–Planck equation, but time and particularly space dependence of diffusivity prevents from getting exact explicit solutions, except for some very elementary cases. Moreover, in structurally disordered media, variations of diffusivity are random, and the need for averaging over random realizations of the disorder makes theoretical analysis particularly challenging. Two typical situations are often investigated. If the disordered medium is immobile (or changes over time scales much longer than that of the diffusion process), the space-dependent diffusivity is considered as a static field, in which diffusion takes place. Whether the diffusivity field is deterministic or random, its spatial profile can significantly impact the diffusive dynamics and, in particular, the distribution of the first-passage time to a reaction event[24–28]. Note that the situation with a random static diffusivity is referred to as "quenched disorder" and enters into a family of models known as "random walks in random environments"[29–34].

In turn, when the medium changes faster than the diffusion time scale, a particle returning to a previously visited point would probe a different local environment that can be modeled by a new realization of random diffusivity at that point. For instance, when a large protein or a vesicle diffuses inside a living cell, other macromolecules, actin filaments, and microtubules can move substantially on comparable time scales, changing the local environment[10,35–37] (see Fig. 1). It is thus natural to consider the diffusivity as a stochastic time-dependent process, $D_t$, referred to as "annealed disorder". The concept of "diffusing diffusivity" was put forward by Chubynsky and Slater[38], and then was further developed by Jain and Sebastian[39,40] and Chechkin et al.[41] (note that the impact of a stochastic volatility onto the distribution of asset returns was investigated much earlier by Drăgulescu and Yakovenko[42]). In ref. [43], we proposed to model the stochastic diffusivity of a particle by a Feller process[44], also known as the square root process or the Cox–Ingersoll–Ross process[45]:

$$dD_t = \frac{1}{\tau}(\bar{D} - D_t)dt + \sigma\sqrt{2D_t}dW_t. \qquad (1)$$

The diffusivity $D_t$ randomly walks around its mean value $\bar{D}$ due to rapid fluctuations of the medium modeled by the standard white noise $dW_t$. The two other parameters of the model characterize the strength of these fluctuations ($\sigma$) and the time scale of medium rearrangements ($\tau$). For a particle moving in the d-dimensional space $\mathbb{R}^d$ free of reactive targets and inert obstacles, we derived the full propagator $P(\mathbf{x}, D, t|\mathbf{x}_0, D_0)$, i.e., the

probability density for a particle started at $\mathbf{x}_0$ with the initial diffusivity $D_0$ to be at $\mathbf{x}$ with the diffusivity $D$ at a later time $t$. When the control dimensionless parameter $\nu = \bar{D}/(\tau\sigma^2)$ is integer, the Feller process (1) is equivalent to the square of an $\nu$-dimensional Ornstein–Uhlenbeck process used for modeling the stochastic diffusivity in ref. [39–41], and our model is thus reduced to the former one. However, integer values of $\nu$ correspond to a weak disorder. In fact, the parameter $1/\nu$ characterizes the disorder strength, i.e., how broad is the distribution of random variations of the diffusivity in a heterogeneous medium. This can be seen by rescaling the diffusivity $D_t$ by $\bar{D}$ and the time $t$ by $\tau$ in Eq. (1), in which case the factor $\sqrt{1/\nu}$ appears in front of the fluctuation term (see Eq. 28 in the Method section). As a consequence, our extension to any real positive $\nu$ and, in particular, to the range $0 < \nu < 1$ that was inaccessible in former works, brought additional features to the annealed model of heterogeneous diffusion.

The above works were devoted to the dynamics itself (MSD scaling, weak ergodicity breaking, non-Gaussian behavior of the propagator, etc.), with no chemical reaction involved. The notable exception is the work by Jain and Sebastian[46], in which the survival probability in crowded rearranging spherical domains was derived. While some first-passage time problems and related reaction kinetics in static disordered media have been addressed[11,13,23,47–49], most former studies were focused on the mean FPT and reaction rates.

In this letter, we couple heterogeneous diffusion to chemical reactions in a medium containing perfectly reactive targets and inert obstacles. We describe a general mathematical framework to translate many results for ordinary homogeneous Brownian motion to heterogeneous diffusion. In particular, we derive

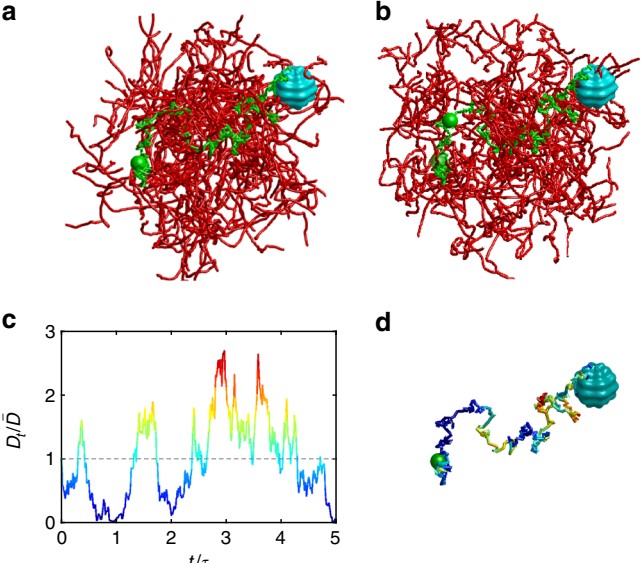

**Fig. 1** Schematic illustration of the annealed disorder model. A dynamic heterogeneous medium is formed by a rearranging polymer solution (red thin tubes mimicking, e.g., actin filaments): (**a**, **b**) two snapshots of a particle (small green ball) diffusing toward a reactive site (light blue bumpy object mimicking, e.g., a protein); a random path (in green) of this particle is added to guide eyes; along the path, the particle interacts with the local environment rearranging on a time scale $\tau$ and thus experiences variable effective diffusivities; (**c**) the environment-induced time-dependent diffusivity $D_t$ is modeled by the Feller process (1); (**d**) once the rearranging environment is taken into account via $D_t$, one deals with the random path from the initial position of the particle (green ball) to the target; the path is explored with a time-dependent "speed" $D_t$, encoded by color as in (**c**)

general spectral decompositions of the full and marginal propagators, the survival probability, the probability density function of the first-passage time to a reaction event, and the macroscopic reaction rate of diffusion-limited reactions. We show how the dynamic disorder broadens the probability density and increases the likelihood of both short and long trajectories to reactive targets. In other words, while the reaction process is slowed down on average, some molecules can reach the reactive targets much faster than via homogeneous diffusion. We discuss biological implications of this counter-intuitive finding, further perspectives and open problems.

## Results

**Heterogeneous diffusion toward reactive targets.** Let us consider a particle diffusing in a fixed volume $\Omega \subset \mathbb{R}^d$ outside an arbitrary configuration of immobile perfectly reactive targets and inert obstacles. The stochastic diffusivity $D_t$, modeled by the Feller process (1), represents the dynamic disorder due to rapid rearrangements of the medium. The particle reacts upon the first encounter with any target and thus disappears, being chemically transformed into another species. This is a standard scheme for most catalytic reactions. In turn, inert obstacles or impenetrable walls just hinder the motion of the particle or confine it in a prescribed spatial region (e.g., inside a living cell). For any bounded domain $\Omega$ (e.g., the cytoplasm confined by the plasma membrane), we obtain the spectral decomposition for the full propagator $P(\mathbf{x}, D, t | \mathbf{x}_0, D_0)$ by solving the Fokker–Planck equation (see Eq. 14 in Method section). As the instantaneous diffusivities $D_0$ and $D$ are hard to access from experiments, we focus throughout this letter on the more common marginal propagator $P(\mathbf{x}, t | \mathbf{x}_0)$, which is obtained by averaging $P(\mathbf{x}, D, t | \mathbf{x}_0, D_0)$ over the diffusivity $D$ at time $t$ and over the initial diffusivity $D_0$ taken from its stationary distribution. We show in the Method section that the propagator $P(\mathbf{x}, t | \mathbf{x}_0)$ admits a general spectral decomposition

$$P(\mathbf{x}, t | \mathbf{x}_0) = \sum_{n=1}^{\infty} u_n(\mathbf{x}) u_n(\mathbf{x}_0) \Upsilon(t; \lambda_n), \qquad (2)$$

where $\lambda_n$ and $u_n$ are the eigenvalues and the $L_2$-normalized eigenfunctions of the Laplace operator $\Delta$ in $\Omega \subset \mathbb{R}^d$, verifying $\Delta u_n + \lambda_n u_n = 0$, subject to absorbing (Dirichlet) and reflecting (Neumann) boundary conditions on the surfaces of targets and obstacles, respectively, and

$$\Upsilon(t; \lambda) = \left( \frac{4\omega e^{-(\omega-1)t/(2\tau)}}{(\omega+1)^2 - (\omega-1)^2 e^{-\omega t/\tau}} \right)^{\nu}, \qquad (3)$$

with $\omega = \sqrt{1 + 4\sigma^2 \tau^2 \lambda}$. Our setting and derivation are much more general than that by Jain and Sebastian who obtained a similar spectral decomposition for a disk with a perfectly reactive boundary for diffusing diffusivity modeled by an $\nu$-dimensional Ornstein–Uhlenbeck process[46]. When either dynamic rearrangements of the medium are too fast ($\tau \to 0$) or its fluctuations are too small ($\sigma \to 0$), the diffusivity is constant, $D_t = \bar{D}$, Eq. (3) is reduced to $\Upsilon_{\text{hom}} = \exp(-\bar{D}t\lambda)$, and one recovers the standard spectral decomposition of the propagator for homogeneous diffusion[50]. While the dynamic disorder is incorporated in Eq. (2) via the explicitly known function $\Upsilon(t; \lambda)$, the structure of the confining domain and its reactive properties are fully "encoded" by the Laplacian eigenmodes, $\lambda_n$ and $u_n$[51]. The function $\Upsilon(t; \lambda_n)$ couples, via the expression for $\omega$, the geometric length scales $\lambda_n^{-1/2}$ of the reactive medium to $\sigma\tau$, which can thus be understood as the disorder length scale.

**First-passage times to a reaction event.** The propagator is the essential ingredient for describing diffusion-limited reactions. In particular, the integral of the propagator $P(\mathbf{x}, t | \mathbf{x}_0)$ over the arrival point $\mathbf{x}$ yields the survival probability of a particle inside the domain, from which the time derivative gives the probability density function of the first-passage time to perfectly reactive targets on the boundary $\partial\Omega$:

$$\rho(t | \mathbf{x}_0) = - \sum_{n=1}^{\infty} u_n(\mathbf{x}_0) \Upsilon'(t; \lambda_n) \int_{\Omega} d\mathbf{x}\, u_n(\mathbf{x}), \qquad (4)$$

where prime denotes the time derivative, $\Upsilon'(t; \lambda) = \frac{\partial}{\partial t} \Upsilon(t; \lambda)$, which is known explicitly from Eq. (3):

$$\Upsilon'(t; \lambda) = - \frac{\nu}{2\tau} \left( \omega - 1 + \frac{2\omega \left(\frac{\omega-1}{\omega+1}\right)^2 e^{-\omega t/\tau}}{1 - \left(\frac{\omega-1}{\omega+1}\right)^2 e^{-\omega t/\tau}} \right) \Upsilon(t; \lambda). \qquad (5)$$

The probability density $\rho(t | \mathbf{x}_0)$ is the likelihood for the reaction event to occur at a given time $t$. Setting appropriate Dirichlet–Neumann boundary conditions, one can describe, for instance, the distribution of the reaction time on catalytic germs in a chemical reactor, or the distribution of the first exit time from a confining domain through "holes" on the boundary (e.g., through water or ion channels on the plasma membrane of a living cell). More generally, this formalism allows one to "translate" many first-passage results known for homogeneous diffusion to heterogeneous one and thus to investigate the impact of the dynamic disorder onto heterogeneous catalysis, diffusive search for multiple targets and escape problems[18,52–56].

When the number of particles is large, multiple reaction events occur at different times, and the overall chemical production can be accurately characterized by the mean FPT or by the macroscopic reaction rate $J(t)$, i.e., the overall flux of diffusing particles onto the reactive target at time $t$. As $\rho(t | \mathbf{x}_0)$ can be interpreted as the probability flux onto the target at time $t$ for a single particle started at $\mathbf{x}_0$ at time 0, the overall flux $J(t)$ is obtained by superimposing these contributions. If there are many independent diffusing particles with a prescribed initial concentration $c_0(\mathbf{x}_0)$, each contribution $\rho(t | \mathbf{x}_0)$ is weighted by the number of particles at $\mathbf{x}_0$ (i.e., by $c_0(\mathbf{x}_0) d\mathbf{x}_0$) that yields

$$J(t) = - \sum_{n=1}^{\infty} \Upsilon'(t; \lambda_n) \int_{\Omega} d\mathbf{x}_0\, c_0(\mathbf{x}_0) u_n(\mathbf{x}_0) \int_{\Omega} d\mathbf{x}\, u_n(\mathbf{x}). \qquad (6)$$

However, many cellular processes are triggered by the arrival of one or few molecules onto the target (e.g., a receptor), and the number of such molecules inside the cell is small. In this case, the mean FPT and the macroscopic rate $J(t)$ are not representative, and the full distribution of the first-passage time is needed[57]. Eq. (4) is thus the crucial step to understand the reaction kinetics in rearranging heterogeneous media. In the following, we focus on the probability density $\rho(t | \mathbf{x}_0)$, bearing in mind straightforward extensions to the reaction rate (its behavior is illustrated in the Method section).

**Fast and slow arrivals to reactive targets.** The probability density $\rho(t | \mathbf{x}_0)$ can span over many orders of magnitude in time so that two reaction times in the same medium can be dramatically different. In order to grasp such a broadness of reaction times, it is instructive to look at reaction events that occur at short and long times after the particle release.

The long-time behavior of the probability density function is determined by the smallest eigenvalue $\lambda_1 > 0$:

$$\rho(t|\mathbf{x}_0) \propto u_1(\mathbf{x}_0)\left(\frac{4\omega_1}{(\omega_1 + 1)^2}\right)^\nu \exp\left(-\frac{2}{1+\omega_1}\bar{D}t\lambda_1\right), \quad (7)$$

with $\omega_1 = \sqrt{1 + 4\sigma^2\tau^2\lambda_1}$. This right tail of the probability density characterizes long trajectories to reactive targets. A diffusing particle fully explores the confining domain and thus almost looses track of the starting point $\mathbf{x}_0$ that affects only a prefactor via the eigenfunction $u_1(\mathbf{x}_0)$. The asymptotic behavior is therefore mainly determined by the eigenvalue $\lambda_1$ which in general exhibits an intricate dependence on the geometry of the confining domain and on the configuration of reactive targets[51,55]. The exponential decay of the probability density function resembles that for homogeneous diffusion with the mean diffusivity $\bar{D}$, but the decay rate is decreased by the factor $(1 + \omega_1)/2 \geq 1$. When the disorder length scale $\sigma\tau$ is much smaller than the largest geometric scale $\lambda_1^{-1/2}$, then $\omega_1 \approx 1$, and one recovers the long-time behavior known for homogeneous diffusion. In this limit, the particle has enough time to probe various diffusivities and to average out the disorder. In the opposite limit of a long-range disorder, $\sigma\tau \gg \lambda_1^{-1/2}$, the decay rate in the exponential function is greatly reduced by the factor $\sigma\tau\sqrt{\lambda_1} \gg 1$, and thus the right tail of the probability density is increased. In particular, the mean FPT to a reactive target, which is essentially determined by the exponential tail, is increased by the factor $\sigma\tau\sqrt{\lambda_1}$. We conclude that the dynamic disorder slows down, on average, the reaction kinetics and search by a single particle.

The short-time behavior of the probability density function of the first-passage time to a perfectly reactive region $\Gamma$ of the boundary is deduced from Eq. 41 of the Method section:

$$\rho(t|\mathbf{x}_0) \propto t^{-1}\left(\delta\sqrt{\nu/(\bar{D}t)}\right)^\nu \exp\left(-\delta\sqrt{\nu/(\bar{D}t)}\right), \quad (8)$$

where $\delta$ is the distance between the starting point $\mathbf{x}_0$ and the reactive region $\Gamma$. This relation, which is valid for $t \ll \min\{\tau, \nu\delta^2/\bar{D}\}$, characterizes short, almost direct trajectories to reactive targets, along which the diffusivity remained almost constant. Looking at the argument of the exponential function in Eq. (8), one can appreciate a dramatic effect of heterogeneous diffusion at short times; in particular, the decay of the probability density function for homogeneous diffusion is much faster:

$$\rho_{\text{hom}}(t|\mathbf{x}_0) \propto t^{-1}\left(\delta/\sqrt{\bar{D}t}\right)\exp\left(-\delta^2/(4\bar{D}t)\right). \quad (9)$$

As a consequence, rapid arrivals of a particle to the reactive region are much more probable for heterogeneous diffusion. In other words, the dynamic character of the disorder allows for larger diffusivities and is thus beneficial for a faster arrival to the target by a single particle, in spite the longer mean FPT. The most probable first-passage time, at which the probability density function reaches its maximum, $(\partial\rho(t|\mathbf{x}_0)/\partial t)|_{t_{\text{mp}}} = 0$, can be estimated from Eq. (8) as $t_{\text{mp}} \approx (1 + 5/(2\nu))^{-1}\delta^2/\bar{D}$. As expected, the most probable FPT is proportional to $\delta^2/\bar{D}$ as for Brownian motion, but the prefactor is controlled by the disorder strength $1/\nu$. In particular, the most probable FPT goes to 0 as the disorder strength $1/\nu$ increases. The distance to the target, $\delta$, is the only relevant geometric length in the short-time regime, which is thus very sensitive to the starting point $\mathbf{x}_0$.

**Respective roles of the disorder strength and scale**. While the above asymptotic relations are universal, the functional form of

the probability density $\rho(t|\mathbf{x}_0)$ depends on the shape of the confining domain and its reactive properties. In spite of intensive studies over the past decades[4,13–15,52,53,58–60], the strong impact of the geometric complexity onto first-passage times and chemical reactions is not fully understood even for homogeneous diffusion. In order to decouple the geometric aspects from the dynamic disorder, we consider as an illustrative example heterogeneous diffusion in a simple yet emblematic domain—a ball. This is a very common model of confinement that was in the scope of many former theoretical studies. Since the radius $R$ of the ball is the only geometric scale of the domain, one can focus exclusively on the impact of the dynamic disorder. The substitution of the explicit form of Laplacian eigenmodes[51,61] into Eq. (4) yields

$$\rho(t|\mathbf{x}_0) = 2\sum_{n=1}^{\infty}(-1)^n \underbrace{\frac{\sin(\pi n\|\mathbf{x}_0\|/R)}{\pi n\|\mathbf{x}_0\|/R}}_{=1 \text{ if } \mathbf{x}_0=0}\Upsilon'(t; \pi^2 n^2/R^2), \quad (10)$$

where $\|\mathbf{x}_0\|$ is the radial coordinate of the starting point $\mathbf{x}_0$. Moreover, we consider the first-passage time to the boundary of a ball from its center, $\mathbf{x}_0 = \mathbf{0}$, to fix the distance to the target: $\delta = R$. Given that the parameters $R$ and $R^2/\bar{D}$ fix the length and time scales for homogeneous diffusion, we investigate the impact of the two other parameters of the dynamic disorder. The asymptotic relations (7) and (8) suggest that the proper dimensionless parameters of the model are the disorder scale $\mu = \sigma\tau/R$ (which compares the spatial extent of the disorder to the size of the domain), and the disorder strength $1/\nu = \tau\sigma^2/\bar{D}$.

Figure 2 compares the exact solution (10) for heterogeneous diffusion, with $\Upsilon'(t;\lambda)$ from Eq. (5), and for homogeneous diffusion with mean diffusivity $\bar{D}$ and $\Upsilon'_{\text{hom}}(t;\lambda) = -\bar{D}\lambda\exp(-\bar{D}t\lambda)$. We explore the parameters space $(\mu, 1/\nu)$ in two complementary ways. In the top panels (a–c), we fix three values of the disorder scale $\mu$ ($10^{-1}$, 1, and 10) and range "continuously" the disorder strength $1/\nu$ from $10^{-1}$ to $10^1$. When the disorder scale is small ($\mu = 0.1$), the particle travels enough distance to the reactive boundary to average out stochastic diffusivities. As a consequence, the long-time behavior of the probability density (its right tail) is close to that of homogeneous diffusion with the mean $\bar{D}$, regardless the disorder strength $1/\nu$ in the considered range. At larger disorder scales ($\mu = 1$ and $\mu = 10$), deviations from homogeneous diffusion at long times become progressively stronger. An increase of the disorder strength $1/\nu$ leads to progressive broadening of the distribution. In particular, the short-time tail of the probability density function is shifted to the left, increasing thus chances of reaching the target at short times. In contrast, the short-time behavior remains almost unaffected by the scale $\mu$, when $\mu$ is not too small (compare cases $\mu = 1$ and $\mu = 10$). This is more clearly seen in the bottom panels (d–f), which show $\rho(t|0)$ for three fixed values $1/\nu$ ($10^{-1}$, 1, and 10) and numerous values of $\mu$ ranging from $10^{-1}$ to $10^1$. The left short-time tail is almost independent of $\mu$ and controlled by $1/\nu$, in agreement with the short-time asymptotic relation (8). In turn, the right tail is affected by both $\mu$ and $\nu$, see Eq. (7). As the disorder weakens ($1/\nu \to 0$ with fixed $\mu$), the probability density $\rho(t|0)$ approaches $\rho_{\text{hom}}(t|0)$ for homogeneous diffusion. In turn, the short-time tail of $\rho(t|0)$ exhibits deviations from $\rho_{\text{hom}}(t|0)$ in the other limit $\mu \to 0$ (with fixed $\nu$), as discussed around Eq. 48 of the Method section.

## Discussion

The discovered broadening of the distribution and increase of its both short- and long-time tails by dynamic disorder are generic and valid for bounded domains beyond balls. Moreover, our study can be extended to unbounded domains, for which the

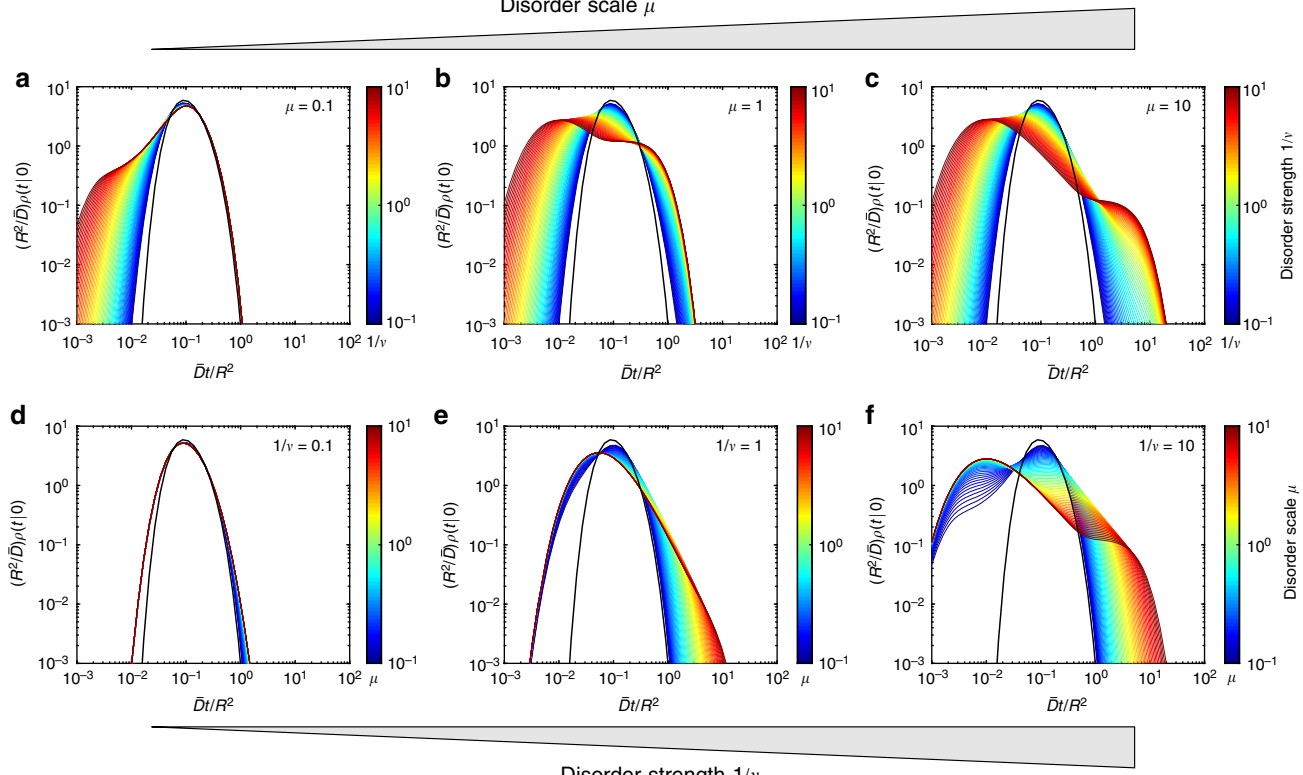

**Fig. 2** The impact of dynamic disorder onto the distribution of first-passage times. The probability density function $\rho(t|0)$ of the first-passage time from the center to the perfectly reactive boundary of a ball of radius $R$ is shown by colored curves for various combinations of dimensionless parameters ($\mu$, $1/\nu$) characterizing the disorder scale and strength: $\mu = \sigma\tau/R$ and $1/\nu = \tau\sigma^2/\bar{D}$. Thick black curve presents $\rho_{hom}(t|0)$ for homogeneous diffusion with diffusivity $\bar{D}$. **a–c** Three values of the disorder scale $\mu$ (0.1 (**a**); 1 (**b**); and 10 (**c**)) and 64 values of the disorder strength $1/\nu$ in the logarithmic range between $10^{-1}$ and $10^1$. **d–f** Three values of the disorder strength $1/\nu$ (0.1 (**d**); 1 (**e**); and 10 (**f**)) and 64 values of the disorder scale $\mu$ in the logarithmic range between $10^{-1}$ and $10^1$. Curves encoded by color, ranging from dark blue ($10^{-1}$) to dark red ($10^1$), as shown by colorbar

analysis becomes more subtle because the spectrum of the Laplace operator is not discrete anymore. In the Method section "Unbounded domains", we provide the explicit representations of the propagator, the survival probability, the probability density, and the macroscopic reaction rate for two unbounded domains: a half-space with a perfectly reactive hyperplane and the exterior of a perfectly reactive ball. In both cases, we show that the long-time behavior of heterogeneous diffusion approaches that of the homogeneous one: the particle has enough time to average out the dynamic disorder, whatever its length scale $\sigma\tau$ (this is equivalent to $\mu = 0$). In particular, we retrieve the Smoluchowski diffusion-limited reaction rate for a spherical target as time goes to infinity[62], while the approach to this stationary limit is moderately affected by the dynamic disorder.

So far, we investigated the impact of the dynamic disorder onto chemical reactions for the particular model (1) of diffusing diffusivity. But, the derived spectral decompositions (2) and (4) turn out to be much more general and can couple the geometric structure of the reactive confining domain $\Omega$ to an arbitrary model of the dynamic disorder represented via the function $\Upsilon(t;\lambda)$. In fact, there are two independent sources of randomness in the annealed model of heterogeneous diffusion: thermal fluctuations that result from local interactions of the medium with a diffusing particle and drive its stochastic motion, and rapid rearrangements of the medium that change the "amplitude" of the local interactions via the stochastic diffusivity. The Laplacian eigenmodes determine the statistics of all possible random paths of a particle in a homogeneous medium due to thermal fluctuations. In turn, the diffusing diffusivity $D_t$ affects only the "speed" at which the particle moves along a randomly chosen path

(Fig. 1). This is the idea of subordination when the integrated diffusivity, $T_t = \int_0^t dt' D_{t'}$, is considered as the "internal time" of a homogeneous process[41]. If the propagator $P_{hom}(\mathbf{x}, T|\mathbf{x}_0)$ of the homogeneous process with a fixed internal time $T$ is known, then the propagator for the subordinated heterogeneous process, in which $T = T_t$ is a random variable, is obtained by averaging $P_{hom}(\mathbf{x}, T_t|\mathbf{x}_0)$ with the probability density function $Q(t;T)$ of the integrated diffusivity $T_t$:

$$P(\mathbf{x}, t|\mathbf{x}_0) = \int_0^\infty dT \, Q(t;T) P_{hom}(\mathbf{x}, T|\mathbf{x}_0). \quad (11)$$

This relation naturally couples two sources of randomness: thermal fluctuations (determining $P_{hom}(\mathbf{x}, T|\mathbf{x}_0)$) and the dynamic disorder (determining $Q(t;T)$). For a homogeneous diffusion in a bounded medium with reactive targets, $P_{hom}(\mathbf{x}, T|\mathbf{x}_0)$ admits the standard spectral decomposition[50], from which Eq. (2) follows, with

$$\Upsilon(t;\lambda) = \int_0^\infty dT \, e^{-\lambda T} Q(t;T) \quad (12)$$

being the Laplace transform of the probability density function $Q(t;T)$. In the same vein, the subordination form for the first-passage time density $\rho(t|\mathbf{x}_0)$ is reproduced from Eq. 103 of the

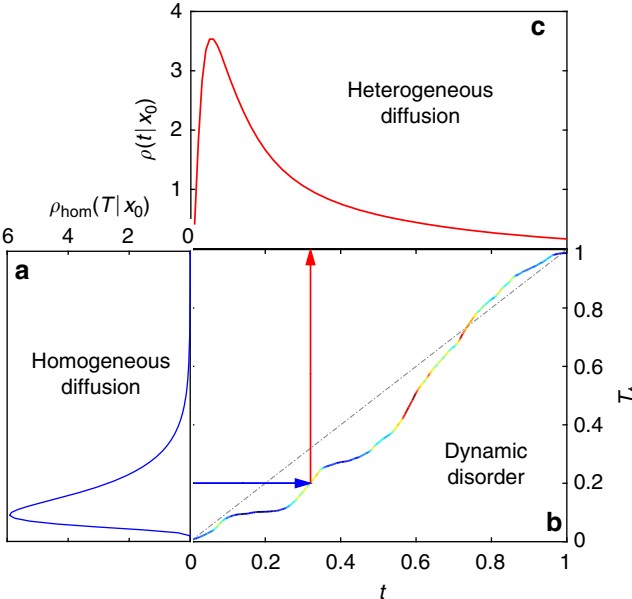

**Fig. 3** Illustration of the subordination concept. The first-passage time to the reactive target is understood as the moment of the first crossing of a random barrier by the integrated diffusivity $T_t$. **a** The geometric structure and reactive properties of the medium determine the probability density function $\rho_{\text{hom}}(T|\mathbf{x}_0)$ of the first-passage time $\mathcal{T}_{\text{hom}}$ to the reactive target by homogeneous diffusion. This FPT is the "duration" of a random Brownian path to the reactive target that sets the barrier to the integrated diffusivity $T_t$ (horizontal blue arrow). The randomness of such Brownian paths results from thermal fluctuations. **b** Rapid rearrangements of the medium lead to a random realization of the "internal time" $T_t$, obtained by integrating the stochastic diffusivity $D_t$ shown in Fig. 1c; colors are reproduced from that figure, ranging from dark blue (low diffusivity) to dark red (high diffusivity). The random moment $t$ (shown by vertical red arrow) when $T_t$ crosses the random barrier $\mathcal{T}_{\text{hom}}$ is the first-passage time to the reactive target by heterogeneous diffusion. **c** The probability density $\rho(t|\mathbf{x}_0)$ of this FPT is obtained by averaging the density $q(t;\mathcal{T}_{\text{hom}})$ over the distribution of $\mathcal{T}_{\text{hom}}$ given by $\rho_{\text{hom}}(T|\mathbf{x}_0)$. The broadening of $\rho(t|\mathbf{x}_0)$ is caused by superimposing two sources of randomness in heterogeneous diffusion: thermal fluctuations (as in $\rho_{\text{hom}}(T|\mathbf{x}_0)$) and medium rearrangements. Arbitrary units are used for this illustrative picture

**Method section:**

$$\rho(t|\mathbf{x}_0) = \int_0^\infty dT \, q(t;T) \rho_{\text{hom}}(T|\mathbf{x}_0), \qquad (13)$$

where $q(t;T)$ is the probability density function of the first-crossing time of a prescribed barrier at $T$ by the integrated diffusivity $T_t$ (the density $q(t;T)$ is also directly related to $Q(t;T)$ and $\Upsilon(t;\lambda)$, see Eqs. 102 and 104 of the Method section). This sub-ordination concept is illustrated by Fig. 3.

In this light, the Feller process (1) for diffusivity $D_t$ can be replaced by another process to reproduce the desired features of dynamic heterogeneous media. In the simplest case when the time-dependent diffusivity $D_t$ is deterministically prescribed, $T_t$ is not random, so that $Q(t;T) = \delta(T - T_t)$ and thus $\Upsilon(t;\lambda) = \exp(-\lambda T_t)$. When the particle undergoes a continuous-time random walk with long stalling periods characterized by an anomalous waiting exponent $0 < \alpha < 1$[12], one gets $\Upsilon(t;\lambda) = E_\alpha(-D_\alpha t^\alpha \lambda)$, where $E_\alpha(z)$ is the Mittag–Leffler function, and $D_\alpha$ is the (constant) generalized diffusion

coefficient[63,64]. One can also consider Lévy-noise-driven processes to model diffusivity with heavy tails[65], geometric Brownian motion to get a nonstationary evolution, or a customized stochastic process to produce the desired distribution of the stationary diffusivity[66]. Once the function $\Upsilon(t;\lambda)$ is computed for the chosen diffusivity model, the coupling to the spatial dynamics of the particle, the related first-passage phenomena, and the consequent reaction kinetics are immediately accessible via the spectral decomposition (2). We stress, however, that the subordination does not provide the full propagator $P(\mathbf{x}, D, t|\mathbf{x}_0, D_0)$, but only the marginal propagator $P(\mathbf{x}, t|\mathbf{x}_0)$.

This letter was focused on diffusion-limited reactions because the related first-passage statistics are essential for characterizing the diffusive transport toward the targets. However, many (bio) chemical reactions involve other "ingredients" such as active transport by motor proteins, bulk reactivity, partially reactive targets, reversible association-dissociation processes and re-binding effects, collective search by multiple particles and the associated (anti-)cooperativity effects, surface diffusion and intermittence, to name but a few. These effects have been progressively incorporated into the theory of homogeneous diffusion-controlled reactions during the past century since the Smoluchowski's seminal paper[62]. Some of these ingredients can be immediately implemented into our formalism. For instance, the Laplace operator governing passive diffusion can be replaced by more general Fokker–Planck operators accounting for an external potential or a drift, allowing one to model active transport in dynamic heterogeneous media such as the cytoplasm of living cells[67]. In turn, the inclusion of some other ingredients remains challenging and requires future investigations. For instance, the macroscopic description of homogeneous diffusion in a medium with partially reactive targets employs the Robin boundary condition that equates the diffusive flux density $-D_0 \frac{\partial}{\partial n} P(\mathbf{x}, t|\mathbf{x}_0)$ toward the target to the reactive flux density $\kappa P(\mathbf{x}, t|\mathbf{x}_0)$ on the target, the reactivity $\kappa$ characterizing the efficiency of reaction (and $\frac{\partial}{\partial n}$ being the normal derivative). An extension of this condition to heterogeneous diffusion with random diffusivity $D_t$ instead of $D_0$ does not seem possible for the marginal propagator $P(\mathbf{x}, t|\mathbf{x}_0)$, and requires considering the full propagator $P(\mathbf{x}, D, t|\mathbf{x}_0, D_0)$. The apparent simplicity of the implementation of the dynamic disorder into the realm of homogeneous diffusions via the function $\Upsilon(t;\lambda)$ is thus deceptive, and the implementation of partial reactivity and some other mechanisms for heterogeneous diffusion raises open mathematical questions.

Another important perspective consists in developing new statistical tools, based on the proposed formalism, to distinguish the impact of the dynamic disorder from other intracellular features (such as visco-elasticity and overcrowding), to identify proper models of diffusing diffusivity from experimental single-particle trajectories, and to infer the parameters of that models. In particular, molecular dynamics simulations could help identifying such models from microscopic principles. In turn, Monte Carlo and finite elements methods allow one to further investigate the role of multiple geometric length scales onto the reaction kinetics in complex geometric confinements.

In summary, we discussed the impact of spatiotemporal disorder of dynamic heterogeneous media onto diffusion-limited reactions, bearing in mind applications to intracellular reactions. A conventional way of tackling such problems would consist in modeling the whole dynamically rearranging medium by means of molecular dynamics simulations. However, a living cell is a very complex system in which a vast number of particles, from water, ions, proteins, actin filaments and microtubules to large organelles such as vesicles and mitochondria, interact to each other, all being confined between the nucleus and the plasma

membrane. Even though molecular dynamics simulations of the intracellular dynamics become more and more accurate and large-scale[35,36], understanding the respective impacts of different cellular mechanisms and processes remains challenging. Theoretical approaches offer a complementary insight by focusing on a particular feature of the intracellular dynamics and ignoring its other aspects. For instance, generalized Langevin equations with memory kernels were invoked to capture visco-elastic properties of the cytoplasm and the related long-time corrections, whereas continuous-time random walk can model molecular caging in an overcrowded environment. Combining such individual mechanisms as elementary pieces, one aims at reconstructing, step by step, the whole mosaic of a cell life. Here, we added a puzzle element by investigating the effects related to dynamic rearrangements of the intracellular medium due to, e.g., actin waves or microtubule movement[68,69]. We greatly simplified the problem by modeling the impact of the medium onto the particle via diffusing diffusivity and thus reducing irrelevant degrees of freedom. The developed theoretical framework revealed that dynamic heterogeneities can actually be beneficial for many biochemical processes in living cells which are triggered by a single molecule[70,71]. More generally, we provided a mathematical ground to advance understanding and modeling of intracellular dynamics to a new level, with potential biomedical and pharmaceutical applications.

## Method

**Derivation of the propagator.** The derivation of the propagator in $\mathbb{R}^d$ from ref. [43] can be generalized to an arbitrary bounded domain $\Omega \subset \mathbb{R}^d$, in which the eigenvalue problem for the Laplace operator is well defined[50,51]. The probability density $P(\mathbf{x}, D, t | \mathbf{x}_0, D_0)$ for a particle started from a point $\mathbf{x}_0$ with the initial diffusivity $D_0$ to be at a point $\mathbf{x}$ with the diffusivity $D$ at a later time $t$ satisfies the forward Fokker–Planck equation in the Itô convention:

$$\frac{\partial P}{\partial t} = \frac{1}{\tau}\frac{\partial}{\partial D}((D - \bar{D})P) + D\Delta P + \sigma^2 \frac{\partial^2}{\partial^2 D}(DP),\quad (14)$$

subject to the initial condition $P(\mathbf{x}, D, t = 0 | \mathbf{x}_0, D_0) = \delta(\mathbf{x} - \mathbf{x}_0)\delta(D - D_0)$ and an appropriate boundary condition on the boundary $\partial\Omega$ of the domain $\Omega$. While the Langevin Eq. (1) automatically ensures the positivity of the diffusivity $D_t$[44,45], the Fokker–Planck equation needs an additional condition at the boundary $D = 0$ in the phase space $(\mathbf{x}, D)$. As discussed in detail in ref. [43], two standard conditions are often employed: the absorbing condition $P(\mathbf{x}, D = 0, t | \mathbf{x}_0, D_0) = 0$ and no flux condition $J_D = -\left(\frac{1}{\tau}(D - \bar{D}) + \sigma^2 \frac{\partial}{\partial D}DP\right)\big|_{D=0} = 0$. The former condition implies that random trajectories in the phase space $(\mathbf{x}, D)$ stop after hitting the boundary $D = 0$: once the diffusivity $D_t$ reaches 0, it gets stuck in this state. As this situation is unphysical, we choose the second condition that ensures the strict positivity of the diffusivity[43,72]. We also impose the regularity condition $P(\mathbf{x}, D, t | \mathbf{x}_0, D_0) \to 0$ as $D \to \infty$.

We sketch the main steps of the derivation. First, one applies the Laplace transform with respect to $D \geq 0$:

$$\tilde{P}(\mathbf{x}, s, t | \mathbf{x}_0, D_0) = \int_0^\infty dD\, e^{-sD} P(\mathbf{x}, D, t | \mathbf{x}_0, D_0),\quad (15)$$

to transform Eq. (14) to

$$\frac{\partial \tilde{P}}{\partial t} + \left(\sigma^2 s^2 + s/\tau + \Delta\right)\frac{\partial}{\partial s}\tilde{P} = -\frac{\bar{D}s}{\tau}\tilde{P},\quad (16)$$

where we used no flux condition at $D = 0$. We decompose $\tilde{P}$ on the complete basis of orthonormal Laplacian eigenfunctions, verifying $\Delta u_n(\mathbf{x}) + \lambda_n u_n(\mathbf{x}) = 0$ in $\Omega$, with the desired boundary condition on $\partial\Omega$, and $\lambda_n$ being the eigenvalues enumerated by $n = 1, 2, \ldots$ in an increasing order. Moreover, the orthogonality of eigenfunctions allows one to search the propagator $\tilde{P}$ in the form

$$\tilde{P}(\mathbf{x}, s, t | \mathbf{x}_0, D_0) = \sum_{n=1}^\infty u_n(\mathbf{x})u_n(\mathbf{x}_0)\tilde{p}(\lambda_n, s, t | D_0).\quad (17)$$

Substitution of this form into Eq. (16) yields a first-order differential equation for the unknown function $\tilde{p}(\lambda, s, t | D_0)$:

$$\frac{\partial \tilde{p}}{\partial t} + \left(\sigma^2 s^2 + s/\tau - \lambda\right)\frac{\partial \tilde{p}}{\partial s} = -\frac{\bar{D}s}{\tau}\tilde{p},\quad (18)$$

subject to the initial condition $\tilde{p}(\lambda, s, t = 0 | D_0) = e^{-sD_0}$. The above equation was solved in ref. [43] by the method of characteristics:

$$\tilde{p}(\lambda, s, t | D_0) = F(D_0, s)e^{-\nu(\omega-1)t/(2\tau)}$$
$$\times \left(\frac{\sigma^2 \tau}{\omega}\left[\left(s + \frac{1+\omega}{2\sigma^2\tau}\right) - \left(s + \frac{1-\omega}{2\sigma^2\tau}\right)e^{-\omega t/\tau}\right]\right)^{-\nu},\quad (19)$$

where

$$F(D_0, s) = \exp\left[\frac{D_0}{2\sigma^2\tau}\left(1 + \omega - \frac{2\omega}{1 - \xi e^{-\omega t/\tau}}\right)\right],\quad (20)$$

$$\xi = 1 - \frac{2\omega}{1 + \omega + 2\sigma^2\tau s},\quad (21)$$

$$\omega = \sqrt{1 + 4\sigma^2\tau^2\lambda}.\quad (22)$$

These relations provide the exact formula (17) for the propagator in the Laplace domain with respect to the diffusivity $D$. The inverse Laplace transform is in general needed to get the propagator $P(\mathbf{x}, D, t | \mathbf{x}_0, D_0)$.

As the diffusivity $D$ at time $t$ is not relevant for most applications, one can focus on the marginal distribution of the position $\mathbf{x}$ by integrating over $D$, which is obtained by setting $s = 0$ in $\tilde{p}(q, s, t | D_0)$:

$$P(\mathbf{x}, t | \mathbf{x}_0, D_0) = \sum_{n=1}^\infty u_n(\mathbf{x})u_n(\mathbf{x}_0)\Upsilon(t; \lambda_n | D_0),\quad (23)$$

where

$$\Upsilon(t; \lambda | D_0) = \left(\frac{2\omega e^{-(\omega-1)t/(2\tau)}}{\omega + 1 + (\omega - 1)e^{-\omega t/\tau}}\right)^\nu$$
$$\times \exp\left(\frac{D_0(\omega+1)}{2\sigma^2\tau}\left(1 - \frac{2\omega}{\omega+1+(\omega-1)e^{-\omega t/\tau}}\right)\right).\quad (24)$$

The marginal distribution (23) is fully explicit in terms of the time dependence.

When the medium is rapidly fluctuating, it is difficult to control the initial diffusivity $D_0$. Since the Feller process (1) for the stochastic diffusivity $D_t$ is stationary, a random "pickup" of the initial diffusivity $D_0$ can be naturally realized by using the stationary distribution of $D_t$, which is known to be the Gamma distribution[43,44]

$$\Pi(D) = \frac{\nu^\nu D^{\nu-1}}{\Gamma(\nu)\bar{D}^\nu}e^{-\nu D/\bar{D}},\quad (25)$$

characterized by the scale $\bar{D}/\nu$ (with the mean $\bar{D}$) and the shape parameter $\nu = \bar{D}/(\tau\sigma^2)$ (with $\Gamma(\nu)$ being the Euler gamma function). We note that, from a physical point of view, local diffusivities should be bounded by the diffusivity of the particle in water, $D_{\max}$, which should thus provide a finite cut-off of the distribution. However, the mean diffusivity $\bar{D}$ in the cytoplasm is much smaller than $D_{\max}$ so that the probability of getting diffusivities larger than $D_{\max}$ is exponentially small. In other words, the exponential decay in Eq. (25) effectively substitutes the finite cut-off.

The $k$th moment of the stationary diffusivity reads

$$\langle D^k \rangle = \begin{cases} \frac{\Gamma(\nu+k)}{\Gamma(\nu)\nu^k}\bar{D}^k & (k > -\nu), \\ \infty & (k \leq -\nu), \end{cases}\quad (26)$$

which is valid even for non-integer and negative $k$. From this relation, one can express the inverse of the shape parameter as

$$\frac{1}{\nu} = \frac{\mathrm{var}\{D\}}{\mathrm{mean}\{D\}^2},\quad (27)$$

i.e., $1/\nu$ characterizes the strength of diffusivity heterogeneity: larger $1/\nu$ corresponds to a broader distribution of stationary diffusivities and thus to stronger disorder. More generally, the role of the parameter $1/\nu$ can be seen by rescaling the diffusivity $D_t$ by its mean $\bar{D}$ and the time $t$ by $\tau$ in Eq. (1) that gives

$$d(D_t/\bar{D}) = (1 - D_t/\bar{D})d(t/\tau) + \sqrt{1/\nu}\sqrt{2D_t/\bar{D}}dW_{t/\tau}.\quad (28)$$

The factor $\sqrt{1/\nu}$ controls the amplitude of the fluctuation term and thus the strength of the dynamic disorder.

The average of the propagator $P(\mathbf{x}, t | \mathbf{x}_0, D_0)$ over random realizations of the initial diffusivity $D_0$, drawn from the Gamma distribution (25), yields the marginal propagator in a bounded domain:

$$P(\mathbf{x}, t | \mathbf{x}_0) = \int_0^\infty dD_0\, \Pi(D_0)P(\mathbf{x}, t | \mathbf{x}_0, D_0).\quad (29)$$

Substitution of Eq. (23) into this relation implies the spectral decomposition (2),

with

$$\Upsilon(t;\lambda) = \int_0^\infty dD_0\, \Pi(D_0)\,\Upsilon(t;\lambda|D_0). \tag{30}$$

The computation of this integral yields Eq. (3).

The reactive properties of the boundary of the confining domain $\Omega$ and its interaction with diffusing particles are introduced via boundary conditions in a standard way[50,52], and fully captured by the Laplacian eigenmodes. When the boundary is a passive, impenetrable wall that constrains the particle inside a bounded domain, Neumann boundary condition is imposed to ensure no probability flux across the boundary: $\partial P/\partial n = 0$, where $\partial/\partial n$ is the normal derivative. In this case, the particle is always present in the domain, and the normalization of the propagator is preserved in time:

$$\int_\Omega d\mathbf{x}\, P(\mathbf{x},t|\mathbf{x}_0) = 1. \tag{31}$$

However, when the boundary contains holes, traps or reactive regions that may kill, adsorb, transfer or transform the particle or modify its state upon the first encounter, Dirichlet boundary condition is imposed on these perfectly reactive parts of the boundary. In this case, the propagator $P(\mathbf{x},t|\mathbf{x}_0)$ should be interpreted as the probability density for a particle started at $\mathbf{x}_0$ to be found at $\mathbf{x}$ at time $t$, without being destroyed or modified on its way. As a consequence, the normalization of the propagator is not preserved and gradually decreases with time. The above integral yields thus the survival probability up to time $t$, $S(t|\mathbf{x}_0)$, for which Eq. (2) implies

$$S(t|\mathbf{x}_0) = \sum_{n=1}^\infty u_n(\mathbf{x}_0)\Upsilon(t;\lambda_n)\int_\Omega d\mathbf{x}\, u_n(\mathbf{x}). \tag{32}$$

This quantity can also be understood as one minus the cumulative distribution function (cdf) of the random first-passage time $\mathcal{T}$, at which the particle reaches the target to be destroyed, chemically transformed or modified on the reactive region: $S(t|\mathbf{x}_0) = 1 - \mathbb{P}_{\mathbf{x}_0}\{\mathcal{T} < t\}$. In other words, $\mathcal{T}$ is the first-passage time to a reaction event, whatever its microscopic mechanism is. The time derivative of the cdf gives the probability density function of this first-passage time:

$$\rho(t|\mathbf{x}_0) = -\sum_{n=1}^\infty u_n(\mathbf{x}_0)\Upsilon'(t;\lambda_n)\int_\Omega d\mathbf{x}\, u_n(\mathbf{x}), \tag{33}$$

where $\Upsilon'(t;\lambda)$, given explicitly by Eq. (5), denotes the time derivative of $\Upsilon(t;\lambda)$ from Eq. (3). Finally, the mean FPT can be obtained by integrating $t\rho(t|\mathbf{x}_0)$ over $t$ from 0 to $\infty$ that yields

$$\langle\mathcal{T}\rangle_{\mathbf{x}_0} = \sum_{n=1}^\infty u_n(\mathbf{x}_0)\int_\Omega d\mathbf{x}\, u_n(\mathbf{x})\int_0^\infty dt\,\Upsilon(t;\lambda_n). \tag{34}$$

Note that the last integral can be expressed in terms of the Gauss hypergeometric function as

$$\int_0^\infty dt\,\Upsilon(t;\lambda) = \frac{2\tau(4\omega)^\nu}{\nu(\omega-1)(\omega+1)^{2\nu}}$$
$$\times\, {}_2F_1\left(\frac{\nu(1-1/\omega)}{2},\nu;\frac{\nu(1-1/\omega)}{2}+1;\frac{(\omega-1)^2}{(\omega+1)^2}\right). \tag{35}$$

It is worth noting that many peculiar properties of heterogeneous diffusion-limited reactions and related first-passage phenomena originate from the average over the initial diffusivity $D_0$. For instance, for a given realization of $D_0$, the short-time behavior of the probability density $\rho(t|\mathbf{x}_0)$ is determined by Eq. (9) for homogeneous diffusion (with $\bar{D}$ replaced by $D_0$). This probability density function is very sensitive to the chosen $D_0$ and exhibits strong variations between random realizations of $D_0$, particularly at short times. This observation naturally raises the question of a reliable interpretation of single-particle realizations and their ensemble average. Moreover, an empirical average over a finite number of realizations depends on that number, and thus may lead to transient regimes. This is particularly clear within the superstatistical approximation (see the following section for details) when the exponential function $\exp(-D_0 t\lambda_n)$ from the spectral decomposition of the propagator for homogeneous diffusion is averaged over $D_0$ drawn from the stationary Gamma distribution (25): while the exact average (over infinitely many realizations) gives a power law $(1+\bar{D}t\lambda_n/\nu)^{-\nu}$, an empirical average over a finite number of realizations yields a linear combination of exponential functions and thus, ultimately, decays exponentially. When the number of realizations increases, this linear combination becomes closer and closer to the power law at intermediate times, but this regime is still terminated by an exponential cut-off. In other words, the ensemble average over the initial diffusivity accurately describes the diffusion-reaction properties of a heterogeneous medium if the number of realizations is large enough.

**Superstatistical approximation**. Although we have derived in the previous section the exact form of the propagator and related quantities, their short-time behavior is determined by infinitely many eigenmodes and thus remains challenging to access. To overcome this difficulty, one can resort to a superstatistical approximation[73,74], a common simplified way for accounting for diffusivity heterogeneities. In a nutshell, the effect of disorder is approximately incorporated by assuming that a particle diffuses with a constant but randomly chosen diffusivity $D_0$, whereas the resulting propagator and related quantities are obtained by averaging over the distribution of the initial diffusivity $D_0$. As discussed in ref. [41,43], the superstatistical description accurately approximates the propagator in $\mathbb{R}$ at short times, $t \ll \tau$, when the stochastic diffusivity $D_t$ does not evolve too far from its initial value $D_0$, but fails at long times. It is instructive to compare this approximation to our exact solution. The propagator for homogeneous diffusion with a constant diffusivity $D_0$ admits a spectral decomposition

$$P_{\text{hom}}(\mathbf{x},t|\mathbf{x}_0) = \sum_{n=1}^\infty u_n(\mathbf{x})u_n(\mathbf{x}_0)\exp(-D_0 t\lambda_n). \tag{36}$$

Since diffusivity heterogeneities in a stationary regime are described by the Gamma distribution (25), the average of the propagator with this distribution yields

$$P_0(\mathbf{x},t|\mathbf{x}_0) = \sum_{n=1}^\infty u_n(\mathbf{x})u_n(\mathbf{x}_0)(1+\lambda_n\bar{D}t/\nu)^{-\nu}, \tag{37}$$

where the subscript 0 highlights the short-time range of validity of this superstatistical approximation. One also approximates the probability density function of the first-passage time as

$$\rho_0(t|\mathbf{x}_0) = \bar{D}\sum_{n=1}^\infty \frac{\lambda_n u_n(\mathbf{x}_0)}{(1+\lambda_n\bar{D}t/\nu)^{\nu+1}}\int_\Omega d\mathbf{x}\, u_n(\mathbf{x}). \tag{38}$$

In particular, the propagator and the probability density function exhibit a power-law long-time decay that disagrees with the exponential decay discussed in the Result section. Nevertheless, we will show below that these superstatistical approximations are accurate at short times.

We focus on the short-time behavior of the probability density function $\rho(t|\mathbf{x}_0)$ of the first-passage time to a perfectly reactive region $\Gamma$ on the boundary of the confining domain $\Omega$. For Brownian motion with diffusivity $D_0$, the short-time behavior of this density is well known:

$$\rho_{\text{hom}}(t|\mathbf{x}_0) \simeq \frac{\delta}{\sqrt{4\pi D_0 t^3}}\exp\left(-\delta^2/(4D_0 t)\right), \tag{39}$$

where $\delta$ is the distance from the starting point $\mathbf{x}_0$ to the reactive region $\Gamma$. As a very fast arrival to the reactive region is realized by a "direct trajectory"[75] from $\mathbf{x}_0$ to the closest points on $\Gamma$, the right-hand side of Eq. (39) is close to the exact probability density function of the first-passage time to an absorbing point on the half-line[52]. The average of Eq. (39) with the Gamma distribution (25) yields the short-time behavior of the probability density function:

$$\rho(t|\mathbf{x}_0) \simeq \frac{2^{1/2-\nu}}{\Gamma(\nu)\sqrt{\pi t}}z_0^{\nu+1/2}K_{\nu-1/2}(z_0), \tag{40}$$

where $z_0 = \delta\sqrt{\nu/(\bar{D}t)}$, and $K_\nu(z)$ is the modified Bessel function of the second kind. As $t \to 0$, one has $z_0 \to \infty$, and the asymptotic behavior of $K_\nu(z)$ yields

$$\rho(t|\mathbf{x}_0) \simeq t^{-1}\frac{2^{-\nu}}{\Gamma(\nu)}\left(\delta\sqrt{\nu/(\bar{D}t)}\right)^\nu e^{-\delta\sqrt{\nu/(\bar{D}t)}}. \tag{41}$$

We note that the numerical prefactor can be affected by the geometric structure of the domain. For instance, if the domain is an interval and the particle starts from the middle, then both absorbing endpoints are equally accessible that doubles chances to reach the target at short times, and the asymptotic relation (41) should be multiplied by 2. Ignoring the numerical prefactor, one gets Eq. (8).

The short-time asymptotic relation (41) is valid as soon as $\delta\sqrt{\nu/(\bar{D}t)} \gg 1$ and $t \ll \tau$ that can be written as

$$t/\tau \ll \min\{1,\delta^2/(\sigma^2\tau^2)\}. \tag{42}$$

When the distance to the target $\delta$ is greater than the disorder length scale $\sigma\tau$, the accuracy of the short-time relation is only limited by the time scale $\tau$. In turn, when $\delta < \sigma\tau$, the major limitation is $t \ll \nu\delta^2/\bar{D}$.

Figure 4 illustrates the quality of the superstatistical approximation of the probability density $\rho(t|0)$ of the first-passage time from the center to the perfectly reactive boundary of a ball of radius $R$. In this case, the superstatistical approximation (38) reads

$$\rho_0(t|0) = \frac{2\pi^2\bar{D}}{R^2}\sum_{n=1}^\infty n^2(-1)^{n-1}\left(1+\frac{\bar{D}t\pi^2 n^2}{\nu R^2}\right)^{-\nu-1}. \tag{43}$$

Note that this superstatistical approximation does not depend on the disorder scale

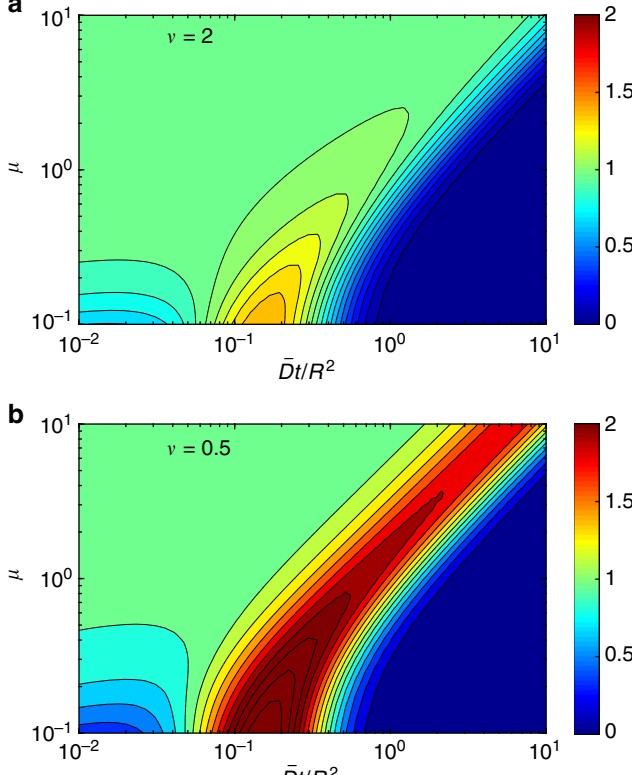

**Fig. 4** The quality of the superstatistical approximation. Illustration for the probability density function of the first-passage time from the center to the perfectly reactive boundary of a ball of radius $R$. The ratio between the exact solution $\rho(t|0)$ from Eq. (10) and its superstatistical approximation $\rho_0(t|0)$ from Eq. (43) is encoded by color and shown by 16 filled contours, for $\nu = 2$ (**a**) and $\nu = 0.5$ (**b**). The approximation is accurate when the ratio is close to 1 (left upper corner). Here $\nu = \bar{D}/(\tau\sigma^2)$ and $\mu = \sigma\tau/R$

$\mu = \sigma\tau/R$. When the disorder scale $\mu$ is large, the diffusivity $D_t$ does not change much from its randomly chosen starting value $D_0$, so that the superstatistical approximation is accurate for a broad range of times (left upper corner of contour plots in Fig. 4). In this regime, deviations appear only at relatively long times (right upper corner). As the disorder scale $\mu$ decreases, the validity range of the superstatistical approximation progressively shrinks toward very short times. This is true for both weak ($1/\nu = 0.5$) and strong ($1/\nu = 2$) disorder, deviations being higher in the latter case. We conclude that the superstatistical approximation and the resulting short-time behavior are accurate when $\mu$ is not too small.

**Limiting behavior of the probability density function**. Figure 2 illustrated the behavior of the probability density function $\rho(t|\mathbf{x}_0)$ of the first-passage time to a perfectly reactive surface of a ball of radius $R$. We explored the space $(\mu, 1/\nu)$ of parameters characterizing the scale and the strength of the dynamic disorder, respectively:

$$\mu = \sigma\tau/R, \quad 1/\nu = \tau\sigma^2/\bar{D}. \quad (44)$$

In particular, we studied the impact of these parameters onto the short-time and long-time tails of the probability density. In this section, we investigate the behavior of the probability density $\rho(t|\mathbf{x}_0)$ in two limits: $\mu \to 0$ (with fixed $1/\nu$) and $1/\nu \to 0$ (with fixed $\mu$).

We first recall that the function $\Upsilon(t;\lambda)$ from Eq. (3) converges to

$$\Upsilon_{\text{hom}}(t;\lambda) = \exp(-\bar{D}t\lambda) \quad (45)$$

for homogeneous diffusion when either the amplitude $\sigma$ of diffusivity fluctuations vanishes (with fixed $\tau$), or their time scale $\tau$ vanishes (with fixed $\sigma$). However, the limiting behavior for $\mu \to 0$ or $1/\nu \to 0$ is more intricate. Fixing $R$ and $\bar{D}$, one can express $\sigma$ and $\tau$ from Eq. (44) as

$$\sigma = \frac{\bar{D}}{R}\frac{1}{\mu\nu}, \quad \tau = \frac{R^2}{\bar{D}}\nu\mu^2. \quad (46)$$

The limit $\mu \to 0$ (with fixed $1/\nu$) implies the double limit $\sigma \to \infty$ and $\tau \to 0$, i.e.,

fluctuations of diffusivity become giant but rapidly reverting to the mean. In turn, the limit $1/\nu \to 0$ (with fixed $\mu$) implies the double limit $\sigma \to 0$ and $\tau \to \infty$, i.e., fluctuations of diffusivity are small but very slowly reverting to the mean. It is thus not clear, *a priori*, whether a diffusing particle would manage to average out such diffusivities to be described by homogeneous diffusion.

To clarify these points, we rewrite the function $\Upsilon(t;\lambda)$ in terms of $\nu$ and $\mu$ as

$$\Upsilon(t;\lambda) = \left(\frac{4\omega\, e^{-(\omega-1)\hat{t}/(2\nu\mu^2)}}{(\omega+1)^2 - (\omega-1)^2 e^{-\omega\hat{t}/(\nu\mu^2)}}\right)^{\nu}, \quad (47)$$

where $\hat{t} = \bar{D}t/R^2$ is the rescaled time, and $\omega = \sqrt{1 + 4\mu^2 R^2\lambda}$.

In the limit $1/\nu \to 0$ (with fixed $\mu$), one can expand the exponential function $e^{-\omega\hat{t}/(\nu\mu^2)}$ in the denominator of Eq. (47) to get, for a fixed $t$,

$$\Upsilon(t;\lambda) \simeq \Upsilon_{\text{hom}}(t;\lambda) + O(1/\nu).$$

As a consequence, the probability density $\rho(t|\mathbf{x}_0)$ approaches that for homogeneous diffusion as $1/\nu$ is getting smaller. Since the time $t$ stands in the small expansion parameter, the functions $\rho(t|\mathbf{x}_0)$ and $\rho_{\text{hom}}(t|\mathbf{x}_0)$ are closer to each other for smaller $t$.

In the other limit $\mu \to 0$ (with $1/\nu$ fixed), one uses the Taylor expansion $\omega \simeq 1 + 2\mu^2 R^2\lambda + O(\mu^4)$ for a fixed $\lambda$ to show that

$$\Upsilon(t;\lambda) \simeq \Upsilon_{\text{hom}}(t;\lambda) + O(\mu^2). \quad (48)$$

One sees again that the function $\Upsilon(t;\lambda)$ converges to that for homogeneous diffusion in this limit. However, the probability density $\rho(t|\mathbf{x}_0)$ remains different from $\rho_{\text{hom}}(t|\mathbf{x}_0)$ at short times. In fact, the expansion (48) holds for any fixed $\lambda$, whereas the spectral decomposition (4) involves terms with Laplacian eigenvalues $\lambda_n$ that grow to infinity as $n$ increases (here we assume that $\lambda_n$ are enumerated in an increasing order). Regardless of the smallness of the parameter $\mu > 0$, there exists an index $n_0$ such that $\mu^2 R^2\lambda_n \gg 1$ for all $n > n_0$ so that the above expansion is not applicable. In other words, for any $\mu > 0$, there remain infinitely many terms $\Upsilon(t;\lambda_n)$ that significantly differ from $\Upsilon_{\text{hom}}(t;\lambda_n)$. As these terms determine the short-time asymptotic behavior of the probability density function, $\rho(t|\mathbf{x}_0)$ exhibits deviations from $\rho_{\text{hom}}(t|\mathbf{x}_0)$ at (very) short times for any $\mu > 0$.

**Unbounded domains**. The derivation of the propagator at the beginning of the Methods section is applicable for any bounded domain $\Omega \subset \mathbb{R}^d$, for which the eigenvalue problem for the Laplace operator is well defined, and the spectrum is known to be discrete. An extension to unbounded domains should handle the continuous spectrum of the Laplace operator, in particular, the absence of $L^2$-normalized eigenfunctions. For instance, the propagator for the whole line $\mathbb{R}$ derived in ref. [43] admits a form similar to Eq. (2), in which the eigenvalues $\lambda_n$ are replaced by $q^2$, the eigenfunctions $u_n(x)$ and $u_n(x_0)$ are replaced by Fourier modes $e^{iqx}$ and $e^{-iqx_0}$, and the sum is turned into the integral over $q$:

$$P_{\mathbb{R}}(x,t|x_0) = \int_{-\infty}^{\infty} \frac{dq}{2\pi} e^{iq(x-x_0)} \Upsilon(t;q^2). \quad (49)$$

To illustrate the impact of dynamic disorder in the case of unbounded domains, we focus on two important examples: the half-space and the exterior of a ball. For these examples, one can use the known form of the propagator in $\mathbb{R}^d$, and apply the image method.

**Half-space**. The propagator in $\mathbb{R}^d$ was derived in ref. [43] in the form

$$P_{\mathbb{R}^d}(\mathbf{x},t|\mathbf{x}_0) = \int_{\mathbb{R}^d} \frac{d\mathbf{q}}{(2\pi)^d} e^{i\mathbf{q}(\mathbf{x}-\mathbf{x}_0)} \Upsilon(t;|\mathbf{q}|^2). \quad (50)$$

In contrast to the Gaussian propagator for homogeneous diffusion, the propagator $P_{\mathbb{R}^d}(\mathbf{x},t|\mathbf{x}_0)$ in $d$ dimensions is not the product of $d$ one-dimensional propagators because $\Upsilon$ is not an exponential function of $|\mathbf{q}|^2$. This is expected because the motions along different directions are correlated via the stochastic diffusivity $D_t$.

The propagator in a half-space $\mathbb{R}^d_+$ with a perfectly reactive hyperplane can be obtained by the image method:

$$P_{\mathbb{R}^d_+}(\mathbf{x},t|\mathbf{x}_0) = P_{\mathbb{R}^d}(\mathbf{x},t|\mathbf{x}_0) - P_{\mathbb{R}^d}(\mathbf{x},t|\widehat{\mathbf{x}}_0), \quad (51)$$

where $\widehat{\mathbf{x}}_0$ is the mirror reflection of $\mathbf{x}_0$ with respect to the reactive hyperplane. The survival probability in the half-space is deduced by integrating this propagator over $\mathbf{x} \in \mathbb{R}^d_+$. Importantly, the statistics of the first-passage time to the reactive hyperplane is not affected by the lateral motion (that is parallel to the reactive hyperplane), as for homogeneous diffusion. In fact, the integral of the propagator $P_{\mathbb{R}^d}(\mathbf{x},t|\mathbf{x}_0)$ over all lateral coordinates yields the one-dimensional propagator in the orthogonal direction (that we choose to be along $x_1$ for clarity):

$$\int_{\mathbb{R}^{d-1}} dx_2 \dots dx_d\, P_{\mathbb{R}^d}(\mathbf{x},t|\mathbf{x}_0) = P_{\mathbb{R}}\left(x_1,t|x_{0,1}\right). \quad (52)$$

In other words, the computation of the survival probability and the probability density function of the first-passage time in the half-space is reduced to that for a half-line with an absorbing endpoint. We focus thus on this one-dimensional problem.

Using the image method, we deduce the propagator on the half-line $(0, \infty)$ with an absorbing endpoint at 0:

$$P(x,t|x_0) = P_{\mathbb{R}}(x,t|x_0) - P_{\mathbb{R}}(x,t|-x_0)$$
$$= -\frac{i}{\pi} \int_{-\infty}^{\infty} dq\, e^{iqx} \sin(qx_0) \Upsilon(t; q^2). \tag{53}$$

This is the probability density for a particle started at $x_0 > 0$ to be at $x \geq 0$ at time $t$, without hitting the absorbing endpoint 0 on its way. Integrating the propagator over the arrival point $x$, one gets the survival probability $S(t|x_0)$

$$S(t|x_0) = \int_0^{\infty} dx\, P(x,t|x_0) = \frac{2}{\pi} \int_0^{\infty} \frac{dq}{q} \sin(qx_0) \Upsilon(t; q^2), \tag{54}$$

from which the probability density function of the first-passage time is

$$\rho(t|x_0) = -\frac{\partial S(t|x_0)}{\partial t} = -\frac{2}{\pi} \int_0^{\infty} \frac{dq}{q} \sin(qx_0) \Upsilon'(t; q^2), \tag{55}$$

where $\Upsilon'(t; \lambda)$ is given by Eq. (5). For comparison, the probability density function for homogeneous diffusion with diffusivity $\bar{D}$ is

$$\rho_{\text{hom}}(t|x_0) = \frac{x_0}{\sqrt{4\pi\bar{D}t^3}} \exp\left(-\frac{x_0^2}{4\bar{D}t}\right). \tag{56}$$

In the long-time limit, the terms $e^{-\omega t/\tau}$ in Eq. (5) vanish, yielding

$$\rho(t|x_0) \simeq \int_0^{\infty} \frac{\nu dq}{\pi\tau q} \sin(qx_0)(\omega-1)e^{-\frac{t}{2}(\omega-1)\frac{t}{\tau}} \left(\frac{4\omega}{(\omega+1)^2}\right)^{\nu}, \tag{57}$$

with $\omega = \sqrt{1 + 4\sigma^2\tau^2 q^2}$. Changing the integration variable and eliminating all terms of order $1/t$ or higher, one gets the classic power-law behavior

$$\rho(t|x_0) \simeq \frac{x_0}{\sqrt{4\pi\bar{D}t^3}} \quad (t \to \infty), \tag{58}$$

that corresponds to Brownian motion (cf. Eq. (56)). In this limit, the particle has enough time to average out the disorder in diffusivities and thus behaves as a Brownian particle with the mean diffusivity $\bar{D}$. This conclusion contrasts with the case of bounded domains, for which the long-time asymptotic behavior could be significantly affected by the disorder (see Eq. (7) and the related discussion in the Results section). The main difference for unbounded domains is the absence of the largest geometric length scale (an analog of $\lambda_1^{-1/2}$) as the Laplacian spectrum is continuous and bounded by zero. From a practical point of view, the long-time behavior is dominated by very long trajectories exploring the unbounded domain so that diffusivity heterogeneities are averaged out independently of their length scale $\sigma\tau$. In particular, the mean FPT is infinite, as for homogeneous diffusion.

The short-time behavior can be obtained via the superstatistical approach by averaging the Gaussian propagator for Brownian motion with the Gamma distribution (25) for diffusivities and then applying the image method. First, one gets the averaged propagator in $\mathbb{R}$

$$P_{\mathbb{R},0}(x,t|x_0) = \frac{\sqrt{\nu}}{\sqrt{\bar{D}t}} \mathcal{K}_{\nu}\left(|x-x_0|\sqrt{\nu/(\bar{D}t)}\right), \tag{59}$$

where we defined

$$\mathcal{K}_{\nu}(z) = \frac{2^{1/2-\nu}}{\Gamma(\nu)\sqrt{\pi}} z^{\nu-1/2} K_{\nu-\frac{1}{2}}(z). \tag{60}$$

The image method yields the averaged propagator on the half-line, from which the superstatistical approximation of the survival probability follows

$$S_0(t|x_0) = 2\int_0^{z_0} dz\, \mathcal{K}_{\nu}(z), \tag{61}$$

where $z_0 = x_0\sqrt{\nu/(\bar{D}t)}$. This integral can be expressed via Struve functions. In turn, the superstatistical approximation of the probability density function is much simpler:

$$\rho_0(t|x_0) = \frac{z_0\,\mathcal{K}_{\nu}(z_0)}{t}. \tag{62}$$

Naturally, we retrieved the right-hand side of Eq. (40), which was obtained as an

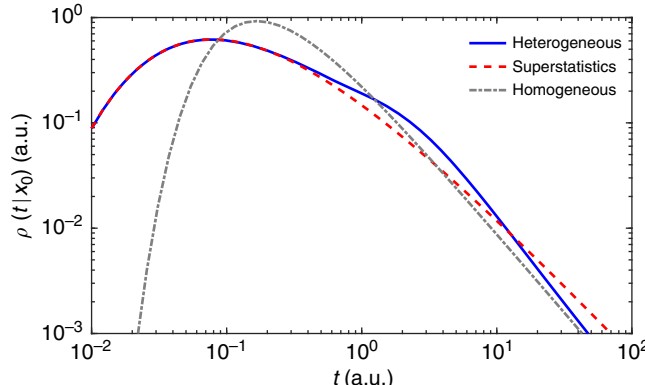

**Fig. 5** First-passage times on the half-line. Probability density function $\rho(t|x_0)$ of the first-passage time to the absorbing endpoint 0 of the half-line $(0, \infty)$, with $\bar{D}=1$, $\sigma=1$, $x_0=1$, and $\tau=2$ (here arbitrary units are used). The exact solution (55) (solid line) is compared to the superstatistical approximation (62) (dashed line) and the probability density function (56) for homogeneous diffusion (dash-dotted line)

approximate asymptotic relation for bounded domains. As $t \to 0$, this relation leads to the asymptotic behavior (8). As expected, the short-time behavior does not depend on the type of the confining domain. It is worth noting that for the half-line, the superstatistical approach captures qualitatively even the long-time asymptotic behavior for $\nu > 1/2$:

$$\rho_0(t|x_0) \simeq \frac{\Gamma(\nu-1/2)\sqrt{\nu}}{\Gamma(\nu)} \frac{x_0}{\sqrt{4\pi\bar{D}t^3}} \quad (t \to \infty), \tag{63}$$

but overestimates the probability density by a numerical factor depending only on $\nu$.

Figure 5 compares the exact probability density $\rho(t|x_0)$ from Eq. (55), its superstatistical approximation (62), and the probability density (56) for homogeneous diffusion. One can see that the superstatistical approximation turns out to be very accurate not only at short times, but also at intermediate times. At long times, this approximation provides the correct power law $t^{-3/2}$ but overestimates the prefactor (cf. Eq. (63)). In turn, the probability density function for Brownian motion yields the correct long-time asymptotic behavior but generally fails.

As discussed in the Results section, the macroscopic reaction rate $J(t)$ (i.e., the diffusive flux onto the reactive target) can be obtained by averaging the probability density $\rho(t|x_0)$ with a prescribed initial concentration of particles $c_0(x_0)$:

$$J(t) = \int_0^{\infty} dx_0\, c_0(x_0)\rho(t|x_0). \tag{64}$$

Setting a uniform initial concentration $c_0$ and using the probability density function in Eq. (55), we get

$$J(t) = -\frac{2c_0}{\pi} \int_0^{\infty} dx_0 \int_0^{\infty} dq\, \frac{\sin(qx_0)}{q} \Upsilon'(t; q^2). \tag{65}$$

To evaluate the integral, we introduce an auxiliary integral

$$I_s(t) = \int_0^{\infty} dx_0\, e^{-sx_0} \int_0^{\infty} dq\, \frac{\sin qx_0}{q} \Upsilon'(t; q^2)$$
$$= \int_0^{\infty} \frac{dq\, \Upsilon'(t; q^2)}{s^2 + q^2} \tag{66}$$

and then get the macroscopic reaction rate

$$J(t) = -\frac{2c_0}{\pi} \lim_{s\to 0} I_s(t) = -\frac{2c_0}{\pi} \int_0^{\infty} \frac{dq\, \Upsilon'(t; q^2)}{q^2}. \tag{67}$$

As expected for one-dimensional setting, the macroscopic reaction rate vanishes in the long-time limit as all diffusing particles are progressively absorbed and finally exhausted.

**Exterior of a ball.** We provide the exact solution to another important example of an unbounded domain – the exterior of a ball of radius $R$ with perfectly reactive boundary. Since the seminal work by Smoluchowski[62], this is an emblematic problem of diffusion-limited reactions.

**Survival probability**. It is convenient to use the representation of the propagator (50) in spherical coordinates derived in ref. [43]:

$$P_{\mathbb{R}^d}(\mathbf{x},t|\mathbf{x}_0) = \frac{\delta^{1-d/2}}{(2\pi)^{d/2}}\int_0^\infty dq\, q^{d/2} J_{\frac{d}{2}-1}(q\delta)\,\Upsilon(t;q^2), \tag{68}$$

where $\delta = \|\mathbf{x}-\mathbf{x}_0\|$ is the distance between the points $\mathbf{x}$ and $\mathbf{x}_0$, and $J_\nu(z)$ is the Bessel function of the first kind.

In three dimensions ($d=3$), representing the points $\mathbf{x}$ and $\mathbf{x}_0$ in spherical coordinates with respect to a fixed center and averaging over the angular coordinates, one can rewrite the above propagator as

$$P_{\mathbb{R}^3}(r,t|r_0) = \int_0^\infty dq\, \frac{\cos(q(r-r_0)) - \cos(q(r+r_0))}{4\pi^2 rr_0}\,\Upsilon(t;q^2) \tag{69}$$

(here we oriented the spherical coordinates in the direction to the point $\mathbf{x}_0$ so that $\mathbf{x}_0 = (r_0,0,0)$ and used thus $\delta = \sqrt{r^2 - 2rr_0\cos\theta + r_0^2}$). Examining this particular form, we realize that the propagator outside a ball of radius $R$ with Dirichlet boundary condition reads

$$P(r,t|r_0) = \int_0^\infty dq\, \frac{\cos(q(r-r_0)) - \cos(q(r+r_0-2R))}{4\pi^2 rr_0}\,\Upsilon(t;q^2). \tag{70}$$

In order to compute the integral over the volume, we first evaluate an auxiliary integral

$$\begin{aligned} I_s(t|r_0) &= 4\pi \int_R^\infty dr\, r\, e^{-sr} P(r,t|r_0) \\ &= \frac{2e^{-sR}}{\pi r_0}\int_0^\infty dq\, \frac{q\sin(q(r_0-R))}{q^2+s^2}\,\Upsilon(t;q^2) \end{aligned} \tag{71}$$

The derivative of this expression with respect to $s$, evaluated at $s=0$ and taken with the sign minus, yields the integral of $P(r,t|r_0)$ over the volume and thus the survival probability:

$$S(t|r_0) = \frac{r_0-R}{r_0} + \frac{2R}{\pi r_0}\int_0^\infty dq\, \frac{\sin(q(r_0-R))}{q}\,\Upsilon(t;q^2). \tag{72}$$

Note that the first term, independent of the function $\Upsilon$, comes from an accurate evaluation of the limit $s\to 0$ of the integral term with $s/(q^2+s^2)^2$ in $\partial I_s(t|r_0)/\partial s$. This term is the probability of escaping to infinity. Note also that $\Upsilon(t=0;q^2)=1$ implies the correct initial condition $S(t=0|r_0)=1$. The time derivative of Eq. (72) yields

$$\rho(t|r_0) = -\frac{2R}{\pi r_0}\int_0^\infty dq\, \frac{\sin(q(r_0-R))}{q}\,\Upsilon'(t;q^2), \tag{73}$$

with $\Upsilon'(t;\lambda)$ given by Eq. (5). As expected, this probability density function is not normalized to 1 because the probability of escape to infinity is not zero.

The long-time asymptotic relation

$$\Upsilon(t;q^2) \simeq \left(\frac{4\omega}{(\omega+1)^2}\right)^\nu \exp\left(-\frac{2\bar{D}tq^2}{1+\omega}\right) \tag{74}$$

implies that the dominant contribution to the integral in Eqs. (70) and (72) comes from $q\approx 0$, at which $\omega\approx 1$, and thus one gets $\Upsilon(t;q^2)\approx \exp(-\bar{D}tq^2)$. As a consequence, the long-time asymptotic behavior of the propagator, of the survival probability, and of the probability density function are close to that for Brownian motion with the constant diffusivity $D=\bar{D}$:

$$P_{\text{hom}}(r,t|r_0) = \frac{\exp\left(-\frac{(r-r_0)^2}{4Dt}\right) - \exp\left(-\frac{(r+r_0-2R)^2}{4Dt}\right)}{8\pi rr_0\sqrt{\pi Dt}}, \tag{75}$$

$$S_{\text{hom}}(t|r_0) = \frac{r_0-R}{r_0} + \frac{R}{r_0}\,\text{erf}\left((r_0-R)/\sqrt{4Dt}\right), \tag{76}$$

and

$$\rho_{\text{hom}}(t|r_0) = \frac{R}{r_0}\frac{(r_0-R)\exp\left(-\frac{(r_0-R)^2}{4Dt}\right)}{\sqrt{4\pi Dt^3}}, \tag{77}$$

where $\text{erf}(z)$ is the error function. As for the half-line, the diffusing particle has enough time to average out heterogeneities of diffusivities and thus to move asymptotically as via homogeneous diffusion.

Applying the superstatistical description to the propagator in Eq. (75) with the Gamma distribution (25) for $D$, one finds

$$\begin{aligned} P_0(r,t|r_0) = \ &\frac{1}{4\pi rr_0}\frac{\sqrt{\nu}}{\sqrt{\bar{D}t}}\Big\{\mathcal{K}_\nu\Big(|r-r_0|\sqrt{\nu/(\bar{D}t)}\Big) \\ &- \mathcal{K}_\nu\Big(|r+r_0-2R|\sqrt{\nu/(\bar{D}t)}\Big)\Big\}, \end{aligned} \tag{78}$$

with $\mathcal{K}_\nu(z)$ defined by Eq. (60). We get thus the superstatistical approximation of the survival probability

$$S_0(t|r_0) = 1 - \frac{2R}{r_0}\int_{z_0}^\infty dz\, \mathcal{K}_\nu(z), \tag{79}$$

with $z_0 = (r_0-R)\sqrt{\nu/(\bar{D}t)}$, and that of the probability density function:

$$\rho_0(t|r_0) = \frac{R}{r_0}\frac{z_0\,\mathcal{K}_\nu(z_0)}{t}. \tag{80}$$

This superstatistical expression provides the short-time asymptotic behavior of the exact probability density function. As expected, this asymptotic relation is almost identical to its one-dimensional counterpart in Eq. (40), apart from the additional factor $R/r_0$ accounting for the probability to reach the target.

Figure 6 illustrates the behavior of the probability density function $\rho(t|r_0)$. As for Fig. 2, we explore various combinations of dimensionless parameters $(\mu, 1/\nu)$ characterizing the disorder scale and strength, in two complementary ways. In the top panels (a–c), we fix three values of the scale $\mu$ and range "continuously" $1/\nu$ from $10^{-1}$ and $10^1$. The short-time behavior of the density $\rho(t|r_0)$ (the left tail) is almost not affected by the scale $\mu$, as expected from the asymptotic relation (8) and the superstatistical approximation (80). In turn, the long-time behavior is mostly determined by $\mu$ but also weakly depends on $\nu$. For the short-range disorder ($\mu = 0.1$), the right tail almost coincides with Eq. (77) for homogeneous diffusion, regardless the value of $1/\nu$ in the considered range. As the scale $\mu$ increases, the particle needs more time to homogenize stochastic diffusivities, and one observes deviations from Eq. (77), which are larger for stronger disorder (larger $1/\nu$). In the bottom panels (d,e,f), we fix three values of the disorder strength $1/\nu$ and change the scale $\mu$ "continuously". One sees again that the left tail is almost independent of $\mu$, while the right tail exhibits such a dependence. We stress that variations of this probability density function are in general lower than that shown in Fig. 2 for a bounded domain. Once again, the exploration of an unbounded domain offers more opportunities for a diffusing particle to homogenize stochastic diffusivities at long times.

**Macroscopic reaction rate**. The macroscopic reaction rate $J(t)$ is obtained by averaging the probability density function in Eq. (73) with a uniform initial concentration $c_0$

$$J(t) = -8c_0 R\int_R^\infty dr_0\, r_0\int_0^\infty dq\, \frac{\sin(q(r_0-R))}{q}\,\Upsilon'(t;q^2). \tag{81}$$

Using again the auxiliary integral (66), we get

$$J(t) = -8c_0 R\lim_{s\to\infty}\int_0^\infty \frac{dq\,\Upsilon'(t;q^2)}{s^2+q^2}\left(R + \frac{2s}{s^2+q^2}\right). \tag{82}$$

Given that $\Upsilon'(t;q^2)\simeq -\bar{D}q^2 + O(q^4)$ as $q\to 0$, the limit of the first term is obtained by setting $s=0$. For the second term, one can extend the integration to $-\infty$ by symmetry and integrate by parts to get

$$\begin{aligned} J(t) = \ &-8c_0 R\bigg\{R\int_0^\infty \frac{dq\,\Upsilon'(t;q^2)}{q^2} \\ &+ \lim_{s\to 0}\frac{1}{2}\int_0^\infty \frac{dq\,s}{s^2+q^2}\frac{\partial}{\partial q}\left(\frac{\Upsilon'(t;q^2)}{q}\right)\bigg\}. \end{aligned}$$

As $s\to 0$, the ratio $s/(s^2+q^2)$ converges to $\pi\delta(q)$ allowing one to evaluate the integral explicitly and yielding

$$J(t) = 4\pi c_0 R\bar{D}\left(1 - \frac{2R}{\pi\bar{D}}\int_0^\infty dq\, \frac{\Upsilon'(t;q^2)}{q^2}\right). \tag{83}$$

For instance, one has $\Upsilon'_{\text{hom}}(t;q^2) = -\bar{D}q^2 e^{-\bar{D}tq^2}$ for Brownian motion with diffusivity $\bar{D}$, and thus retrieves the classic Smoluchowski reaction rate[62]:

$$J_{\text{hom}}(t) = 4\pi c_0 R\bar{D}\left(1 + \frac{R}{\sqrt{\pi\bar{D}t}}\right). \tag{84}$$

In the long-time limit, the second term in both Eqs. (83) and (84) vanishes, and one

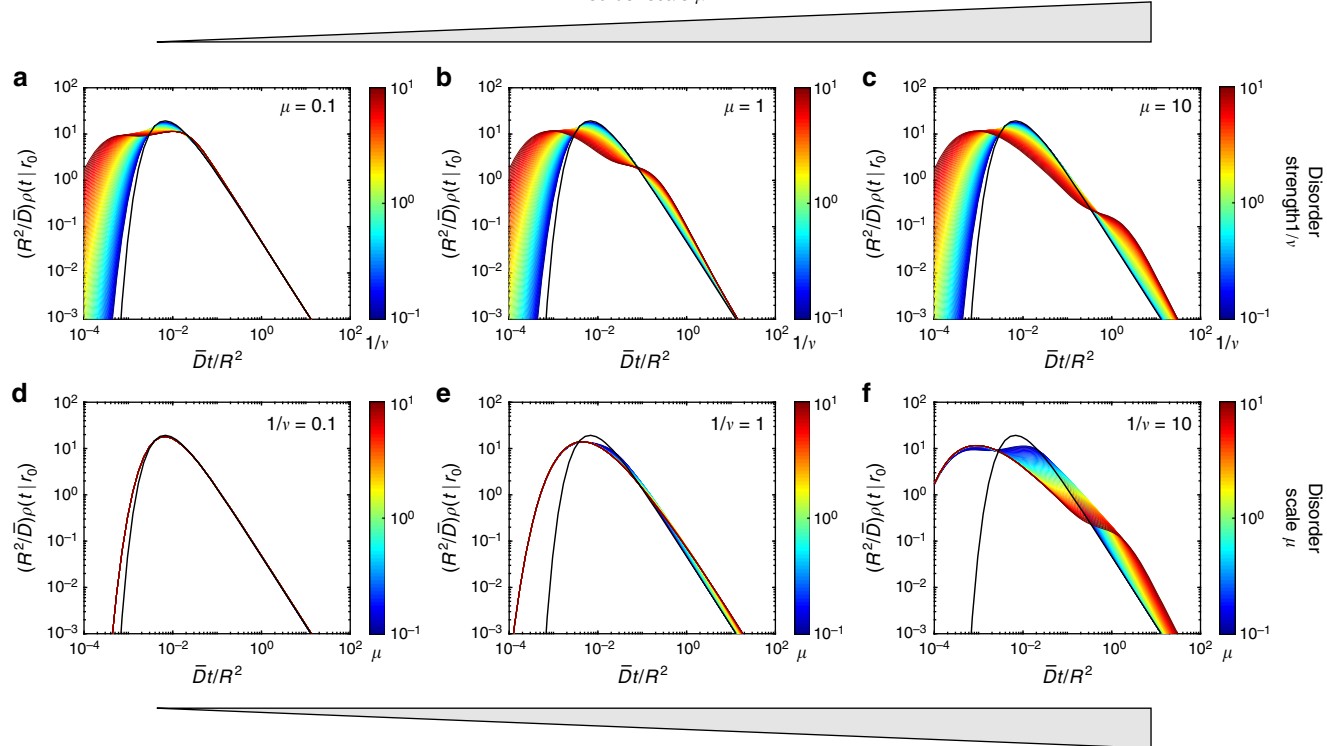

**Fig. 6** The impact of dynamic disorder onto the distribution of first-passage times for the exterior of a ball. The probability density function $\rho(t|r_0)$ of the first-passage time to a perfectly reactive ball of radius $R$ from the exterior space (with $r_0/R = 1.2$) is shown by color lines for various combinations of dimensionless parameters ($\mu$, $1/\nu$) characterizing the disorder scale and strength: $\mu = \sigma\tau/R$ and $1/\nu = \tau\sigma^2/\bar{D}$. Thick black line presents the probability density $\rho_{\mathrm{hom}}(t|r_0)$ from Eq. (77) for homogeneous diffusion with diffusivity $\bar{D}$. **a–c** Three values of the disorder scale $\mu$ (0.1 (**a**); 1 (**b**); and 10 (**c**)) and 64 values of the disorder strength $1/\nu$ in the logarithmic range between $10^{-1}$ and $10^{1}$. **d–f** Three values of the disorder strength $1/\nu$ (0.1 (**d**); 1 (**e**); and 10 (**f**)) and 64 values of the disorder scale $\mu$ in the logarithmic range between $10^{-1}$ and $10^{1}$. Curves encoded by color, ranging from dark blue ($10^{-1}$) to dark red ($10^{1}$), as shown by colorbar

recovers the Smoluchowski steady-state reaction rate:

$$J_S = 4\pi c_0 \bar{D} R. \tag{85}$$

In turn, the approach to the steady-state solution differs for homogeneous and heterogeneous cases.

Using the relation (104), one can rewrite the integral in Eq. (83) as

$$\int_0^\infty dq \frac{\Upsilon'(t; q^2)}{q^2} = -\frac{\sqrt{\pi}}{2} \int_0^\infty dT \frac{q(t; T)}{\sqrt{T}}, \tag{86}$$

where $q(t;T)$ is the probability density function of the first moment $t$ when the integrated diffusivity $T_t$ crosses the level $T$ (see the last section). In the short-time limit, one can resort again to the superstatistical approximation by setting $T_t \simeq Dt$, with $D$ randomly drawn from the Gamma distribution (25). In other words, we approximate $q(t;T)$ as $q(t;T) \approx \langle \delta(t - T/D) \rangle$, where the average is over all random realizations of $D$. This gives the following short-time approximation:

$$q(t;T) \approx \frac{1}{\Gamma(\nu)t} \left( \frac{\nu T}{\bar{D}t} \right)^\nu \exp\left( -\frac{\nu T}{\bar{D}t} \right). \tag{87}$$

Substitution of this approximation into Eq. (86) results in the short-time asymptotic behavior of the rate:

$$J(t) \simeq 4\pi c_0 R\bar{D} \left( 1 + \frac{\Gamma(\nu + 1/2)}{\sqrt{\nu}\,\Gamma(\nu)} \frac{R}{\sqrt{\pi\bar{D}t}} \right) \quad (t \to 0). \tag{88}$$

This relation is close to Eq. (84) for Brownian motion with mean diffusivity $\bar{D}$, in which the divergent $t^{-1/2}$ term is multiplied by the explicit prefactor $\frac{\Gamma(\nu+1/2)}{\sqrt{\nu}\,\Gamma(\nu)}$ depending only on $\nu$. This prefactor monotonously grows from 0 to 1 as $\nu$ increases from 0 to infinity (the limit $\nu \to \infty$ corresponding to Brownian motion). As a consequence, the dynamic disorder tends to diminish the macroscopic reaction rate, in agreement with our statement about an increase of the mean FPT in bounded domains. In turn, the impact of disorder for unbounded domains is rather weak, for instance, the prefactor is 0.8 for $\nu = 1/2$. The approximate asymptotic

relation (88) does not depend on the disorder scale $\mu$. According to the superstatistical approximation, this relation is actually the lower bound for the flux $J(t)$ corresponding to the limit $\tau \to 0$ or, equivalently, $\mu \to \infty$. In turn, the exact expression (84) for Brownian motion corresponds to the limit $\mu \to 0$ and thus is close to the upper bound for $J(t)$. Although the flux is not necessarily a monotonous function of $\mu$, this qualitative analysis accurately describes the behavior of the flux $J(t)$. Figure 7 shows the macroscopic reaction rate $J(t)$ from Eq. (83) normalized by its steady-state value $J_S$. For both weak ($1/\nu = 0.5$) and strong ($1/\nu = 2$) disorder, a substantial increase of the disorder scale $\mu = \sigma\tau/R$ from $10^{-1}$ to $10^{1}$ has only a minor effect, and all curves are close to both the classic flux $J_{\mathrm{hom}}(t)$ from Eq. (84) and the asymptotic relation (88).

We stress that the total flux $J(t)$ was computed by integrating the probability fluxes $\rho(t|\mathbf{x}_0)$. In turn, the common way of obtaining the total flux consists in finding the concentration profile $c(\mathbf{x}, t)$ and then integrating the diffusive flux density, $-D\partial c(\mathbf{x}, t)/\partial n$, over the target surface $\partial\Omega$, where $\partial/\partial n$ is the normal derivative oriented outward the confining domain. However, the diffusivity $D$ is random in the annealed model of heterogeneous diffusion that prohibits using the above form of the diffusive flux density. If the random $D$ is replaced by the mean diffusivity $\bar{D}$, the total flux could then be approximated as

$$\begin{aligned} J_{\mathrm{app}}(t) &= \int_{\partial\Omega} d\mathbf{s} \cdot \left( -\bar{D} \frac{\partial c_0 S(t|\mathbf{x}_0)}{\partial n} \right)\bigg|_{\mathbf{x}_0 = \mathbf{s}} \\ &= 4\pi R c_0 \bar{D} \left( 1 + \frac{2R}{\pi} \int_0^\infty dq\, \Upsilon(t; q^2) \right). \end{aligned} \tag{89}$$

While the long-time limit of the flux (the first term) is the same as in the exact solution (83), the approach to this limit, given by the second term, is different. The formulas (83) and (89) are identical only for homogeneous diffusion. This computation illustrates some pitfalls of applying conventional tools of homogeneous diffusion to heterogeneous one.

For comparison, we also compute the macroscopic reaction rate for heterogeneous diffusion inside a ball of radius $R$. From Eq. (6), one gets

$$J(t) = -\frac{8c_0 R^3}{\pi} \sum_{n=1}^{\infty} \frac{\Upsilon'(t; \pi^2 n^2/R^2)}{n^2}, \tag{90}$$

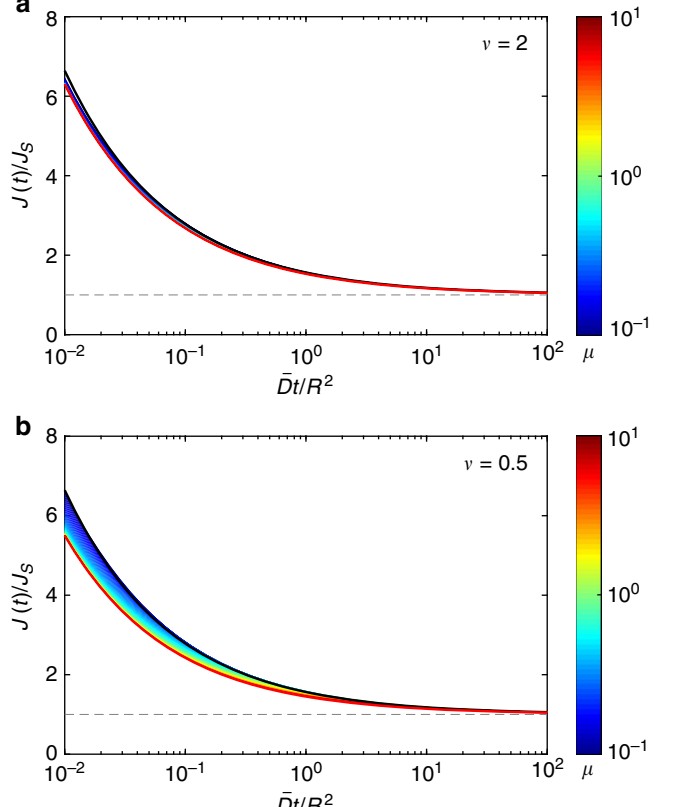

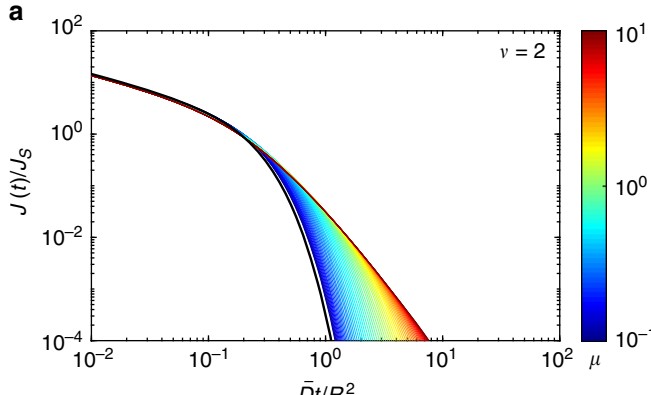

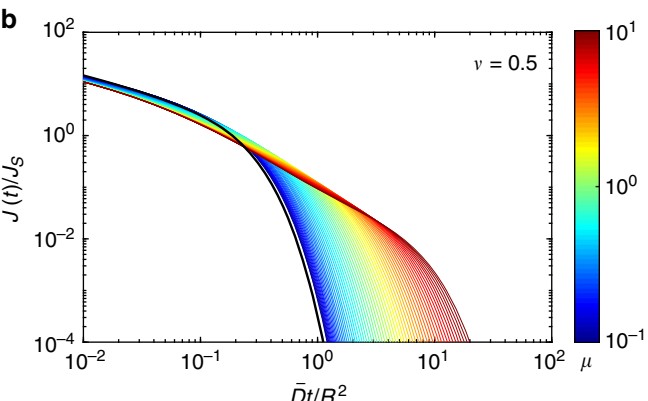

**Fig. 7** The macroscopic reaction rate for the exterior of a ball. The diffusive flux $J(t)$ onto the perfectly reactive boundary of a ball of radius $R$ from Eq. (83), normalized by the steady-state Smoluchowski rate $J_S$ from Eq. (85). The disorder scale $\mu$ takes 64 values in the logarithmic range from $10^{-1}$ (dark blue) to $10^1$ (dark red), as indicated in the colorbar. The shape parameter $\nu$ is set to 2 (**a**) or 0.5 (**b**). The mean diffusivity $\bar{D}$ is fixed, whereas two other parameters of the model are $\tau = R^2\mu^2\nu/\bar{D}$ and $\sigma = \bar{D}/(R\mu\nu)$. The upper black curve presents the flux $J_{\text{hom}}(t)/J_S$ from Eq. (84) for homogeneous diffusion (corresponding to $\mu = 0$), whereas the lower red curve shows the asymptotic relation (88) corresponding to the limit $\mu \to \infty$

**Fig. 8** The macroscopic reaction rate for the interior of a ball. The diffusive flux $J(t)$ onto the perfectly reactive surface of a ball of radius $R$, from Eq. (90), normalized by the Smoluckowski steady-state rate $J_S$ from Eq. (85), with the uniform initial concentration $c_0$ inside the ball, $\nu = 2$ (**a**) and $\nu = 0.5$ (**b**) and 64 values of $\mu$ in the logarithmic range between $10^{-1}$ and $10^1$ (curves encoded by color, ranging from dark blue to dark red, as shown by colorbar). Thick black line shows the macroscopic reaction rate for homogeneous diffusion with diffusivity $\bar{D}$

where $c_0$ is the uniform initial concentration. Figure 8 shows the behavior of this rate, normalized for convenience by the Smoluchowski steady-state rate $J_S$ from Eq. (85). As this confining domain is bounded, the reaction rate vanishes at long times, as all particles will finally react. This is an evident difference from Fig. 7, in which the reaction rate reaches a nonzero limit $J_S$. For a fixed disorder strength $1/\nu$, the curves exhibit a much stronger dependence on the disorder scale $\mu$ for interior diffusion than for exterior one. This observation re-confirms that the dynamic disorder is averaged more efficiently in unbounded domains. In turn, one observes in both Figs. 7, 8 a broader dispersion of curves for stronger disorder. Finally, one sees that the dynamic disorder leads to a higher reaction rate at long times, in agreement with our conclusion that the reaction kinetics is slowed down on average and thus more particles remain present in the confining domain.

**Collective search by multiple independent particles**. The description of a single particle opens a way to investigate some basic multiparticle effects. For instance, when $N$ independent particles simultaneously search for a target, the distribution of the first arrival is still determined by the survival probability for a single particle[76,77]. If the starting points of these particles are uniformly distributed in a region $\Omega = \{\mathbf{x} \in \mathbb{R}^3 : R < \|\mathbf{x}\| < R_{\text{max}}\}$ around the spherical target of radius $R$, one gets

$$S_N(t) = \mathbb{P}\{\min\{\mathcal{T}_1, \ldots, \mathcal{T}_N\} > t\} = \left(\int_\Omega \frac{d\mathbf{x}_0}{V} S(t|\mathbf{x}_0)\right)^N, \quad (91)$$

where $V$ is the volume of $\Omega$, and $\mathcal{T}_1, \ldots, \mathcal{T}_N$ are independent first-passage times for $N$ particles. In the thermodynamic limit, when both $N$ and $V$ (or $R_{\text{max}}$) tend to

infinity but the density $c_0 = N/V$ remains fixed, one finds

$$-\ln(S_\infty(t)) = c_0 \int_{\|\mathbf{x}_0\| > R} d\mathbf{x}_0 (1 - S(t|\mathbf{x}_0)). \quad (92)$$

The right-hand side is the number of particles that reacted up to time $t$ which can also be obtained by integrating the flux $J(t)$ from Eq. (83):

$$S_\infty(t) = \exp\left(-\int_0^t dt' J(t')\right)$$
$$= \exp\left(-4\pi c_0 R\bar{D}\left(t + \frac{2R}{\pi\bar{D}}\int_0^\infty dq \frac{1 - \Upsilon(t;q^2)}{q^2}\right)\right). \quad (93)$$

Once again, this quantity is fully determined by $\Upsilon(t;\lambda)$. In particular, one can use the above asymptotic relations to study the behavior of $S_\infty(t)$ at short and long times.

We note, however, that the validity of the assumption of independent particles is debatable in the context of dynamically rearranging media. In fact, when two particles come close to each other, they probe the same local environment and thus should have similar diffusivities. As a consequence, the stochastic diffusivities of these particles become correlated (locally in time). The impact of this intricate correlation mechanism remains an open challenging problem for future investigations. We also note that the same issue concerns the macroscopic reaction rate $J(t)$ in Eq. (6), which is obtained by superimposing probability fluxes $\rho(t|\mathbf{x}_0)$ from independent particles with a prescribed initial concentration $c_0(\mathbf{x}_0)$.

**Subordination approach**. The subordination approach consists in treating time-dependent diffusivity as changing the "internal time" of the process[41]. When the diffusivity $D_t$ is deterministic, the diffusion equation for the propagator, $\partial P/\partial t =$

$D_t \Delta P$, can be reduced to the "canonical" form, $\partial P_{\text{hom}}/\partial T_t = \Delta P_{\text{hom}}$ with unit diffusivity, where a new "internal time" variable is

$$T_t = \int_0^t dt' \, D_{t'} \tag{94}$$

(we call $T_t$ "internal time", in spite of its units m$^2$; to get usual time units, one can divide it by $\bar{D}$ or another diffusivity). In other words, time-dependent diffusion can be understood as traveling along a random path generated by ordinary Brownian motion, but with a variable, time-dependent "speed" $dT_t/dt = D_t$.

The same argument holds for stochastic diffusivity $D_t$, in which case the internal time $T_t$ is a stochastic process. The conventional spectral decomposition of the propagator in the internal time $T_t$,

$$P_{\text{hom}}(\mathbf{x}, T_t | \mathbf{x}_0) = \sum_{n=1}^{\infty} u_n(\mathbf{x}) u_n(\mathbf{x}_0) e^{-\lambda_n T_t}, \tag{95}$$

should be averaged with the probability density function $Q(t;T)$ of the integrated diffusivity $T_t$:

$$\begin{aligned} P(\mathbf{x}, t | \mathbf{x}_0) &= \langle P_{\text{hom}}(\mathbf{x}, T_t | \mathbf{x}_0) \rangle_{T_t} \\ &= \int_0^{\infty} dT \, Q(t;T) \, P_{\text{hom}}(\mathbf{x}, T | \mathbf{x}_0) \\ &= \sum_{n=1}^{\infty} u_n(\mathbf{x}) u_n(\mathbf{x}_0) \underbrace{\int_0^{\infty} dT \, Q(t;T) e^{-\lambda_n T}}_{\Upsilon(t;\lambda_n)}. \end{aligned} \tag{96}$$

One gets therefore the natural interpretation (12) of $\Upsilon(t;\lambda)$ as the Laplace transform of the probability density function $Q(t;T)$ of the integrated diffusivity $T_t$. The related first-passage times for the Feller process were investigated[78]. The probability density function $Q(t;T)$ "translates" the internal time $T$ into the physical time $t$. If $D_t$ is deterministic, then $Q(t;T) = \delta(T - T_t)$ and thus $\Upsilon(t;\lambda) = \exp(-\lambda T_t)$, as expected. Note that here we considered the internal time $T_t$ averaged over the initial diffusivity $D_0$ drawn from the stationary Gamma distribution (25). In turn, if $D_0$ is fixed, the function $\Upsilon(t;\lambda)$ is replaced by $\Upsilon(t;\lambda|D_0)$ from Eq. (24), which is the Laplace transform of the corresponding probability density function $Q(t;T|D_0)$.

The numerical computation of the probability density function $Q(t;T)$ would require the inversion of the Laplace transform. In turn, the moments of the integrated diffusivity $T_t$ can be easily obtained via Eq. (12):

$$\left\langle (T_t)^k \right\rangle = (-1)^k \lim_{\lambda \to 0} \frac{\partial^k \Upsilon(t;\lambda)}{\partial \lambda^k}. \tag{97}$$

In particular, the mean and the variance are

$$\langle T_t \rangle = \bar{D} t, \quad \text{var}\{T_t\} = \frac{2\tau^2 \bar{D}^2}{\nu} \left( t/\tau - 1 + e^{-t/\tau} \right). \tag{98}$$

As expected, the mean integrated diffusivities for heterogeneous and homogeneous diffusions are identical and grow linearly with time. The variance grows quadratically at small times ($t \ll \tau$) and linearly at large times ($t \gg \tau$). As a consequence, the squared coefficient of variation,

$$\frac{\text{var}\{T_t\}}{\langle T_t \rangle^2} = \frac{2\tau}{\nu t} \left( 1 - \frac{\tau}{t} \left( 1 - e^{-t/\tau} \right) \right), \tag{99}$$

monotonously decreases from $1/\nu$ at $t = 0$ to zero as $t \to \infty$. The shape parameter $\nu$ thus controls the broadness of the distribution of $T_t$ at short times, given that the initial diffusivity $D_0$ is randomly picked up from the Gamma distribution (25), see Eq. (27). Note also that the right-hand side of Eq. (99) coincides with the non-Gaussian parameter for the one-dimensional heterogeneous diffusion in the free space $\mathbb{R}$[43].

In the same vein, the first-passage time $\mathcal{T}$ to a reactive target can be related to the first-crossing time of a random barrier by the stochastic process $T_t$ (Fig. 3). In fact, one can first generate a random path to the target by ordinary Brownian motion with unit diffusivity and then consider a particle traveling along this path with a time-dependent "speed". The target is reached when the whole path is passed through, i.e., when the internal time $T_t$ attains the duration $\mathcal{T}_{\text{hom}}$ of the Brownian path, which is random and determined by the conventional probability density function of the first-passage time for Brownian motion with unit diffusivity

$$\begin{aligned} \rho_{\text{hom}}(T | \mathbf{x}_0) &= \frac{\mathbb{P}_{\mathbf{x}_0}\{\mathcal{T}_{\text{hom}} \in (T, T+dT)\}}{dT} \\ &= \sum_{n=1}^{\infty} u_n(\mathbf{x}_0) \lambda_n e^{-\lambda_n T} \int_{\Omega} d\mathbf{x} \, u_n(\mathbf{x}). \end{aligned} \tag{100}$$

Let $\delta_T = \inf\{t > 0 : T_t > T\}$ be the random time when the process $T_t$ crosses a fixed

level $T$. Since $T_t$ monotonously increases, one has

$$\mathbb{P}\{\delta_T > t\} = \mathbb{P}\{T_t < T\} = \int_0^T dT' \, Q(t; T'). \tag{101}$$

In particular, the probability density function of the random time $\delta_T$ reads

$$q(t; T) = -\frac{\partial \mathbb{P}\{\delta_T > t\}}{\partial t} = -\int_0^T dT' \frac{\partial Q(t; T')}{\partial t}. \tag{102}$$

If now the level $T$ is the random duration of the Brownian path, $T = \mathcal{T}_{\text{hom}}$, the random time $\mathcal{T} = \delta_{\mathcal{T}_{\text{hom}}}$ is the first-passage time to the reactive target, and its probability density function is obtained by averaging $q(t; \mathcal{T}_{\text{hom}})$ over the distribution of $\mathcal{T}_{\text{hom}}$:

$$\rho(t | \mathbf{x}_0) = \int_0^{\infty} dT \, q(t; T) \rho_{\text{hom}}(T | \mathbf{x}_0). \tag{103}$$

Substitution of Eq. (100) into this relation allows one to retrieve Eq. (4). Multiplying Eq. (102) by $e^{-\lambda T}$ and integrating over $T$ from 0 to infinity, one can express the Laplace transform of the density $q(t;T)$ as

$$\int_0^{\infty} dT \, e^{-\lambda T} q(t; T) = -\frac{\Upsilon'(t; \lambda)}{\lambda}, \tag{104}$$

where Eq. (12) was used. One can see that the function $\Upsilon(t;\lambda)$ and its time derivative $\Upsilon'(t;\lambda)$, explicitly known from Eqs. (3) and (5), fully determine the densities $Q(t;T)$ and $q(t;T)$ via Laplace transforms.

In summary, Eqs. (96) and (103) couple the spatial aspects of the problem (such as the geometric structure of the medium, the shape, location and reactivity of the targets, and the starting point) to the dynamic disorder represented by the stochastic diffusivity. The spatial features do not depend on diffusivity and are determined by homogeneous diffusion (ordinary Brownian motion). In turn, the disorder aspects are captured via the distribution of the integrated diffusivity $T_t$. Although we focused on the stochastic diffusivity modeled by the Feller process (1), one can explore other models such as, e.g., reflected Brownian motion[38], Lévy-driven stochastic diffusivity[65], or geometric Brownian motion.

On the other hand, the subordination approach is limited to the marginal propagator $P(\mathbf{x}, t | \mathbf{x}_0)$ and related quantities (such as the probability density function $\rho(t | \mathbf{x}_0)$) and does not yield the full propagator $P(\mathbf{x}, D, t | \mathbf{x}_0, D_0)$ that we obtained in the first part of Method section by solving the Fokker–Planck equation. We also stress that the subordination does not resolve the problem of partially reactive targets with Robin boundary condition, as discussed in the Discussion section. In fact, even though the conventional spectral decomposition (95) is valid for partially reactive boundary condition, its formal extension to heterogeneous diffusion via the subordination (96) remains debatable as it would correspond to a modified model of stochastic diffusivity $D_t$, in which $D_t$ should take a fixed prescribed value when the particle is on the target. A proper description of heterogeneous diffusion toward partially reactive targets remains an open mathematical problem.

**Code availability**. All figures have been prepared by means of Matlab software. The plotted quantities have been computed by explicit formulas provided in the letter by using custom routines for Matlab software. While the explicit form makes these numerical computations straightforward, custom routines are available from the corresponding author upon request.

## Data availability

Data sharing not applicable to this article as no datasets were generated or analyzed during the current study.

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

## Acknowledgments

D.S.G. acknowledges the support under Grant no. ANR-13-JSV5-0006-01 of the French National Research Agency.

## Author contributions

Y.L. and D.S.G. designed research; Y.L., N.M., and D.S.G. performed research and analyzed the results; D.S.G. wrote the paper.

## Additional information

**Competing interests:** The authors declare no competing interests.

