## [Peer Review File · Nature Communications]

Reviewers' comments:

Reviewer #1 (Remarks to the Author):

Nature Communications manuscript NCOMMS-18-14269-T
"Diffusion-controlled reactions in dynamic heterogeneous media"
Dr Grebenkov and colleagues

This manuscript treats a basic feature of diffusion-controlled reactions--namely, its first-passage characteristics---when the particle diffusion coefficient is itself time dependent in a stochastic matter, as embodied by Eq. (1). This is a high-quality manuscript in which a technically challenging problem---diffusive motion with a fluctuating diffusion coefficient---is solved. The manuscript should be published, and preferably in a high-profile journal.

However, in its present form, I'm not convinced that it is suitable for Nature Communications.

My first concern is the overlap of the present manuscript with Ref. [50]. It appears that many of the technical details in the manuscript and the supplementary material were first derived in Ref. [50]. The authors need to clearly state what is new beyond Ref. [50] in the present manuscript.

A second issue is the non self-contained nature of the manuscript. In the text, there is reference to subordination and to superstatistics, both of which are treated in the SI. However, the meaning of these two concepts in the context of the new manuscript results is not given, as far as I can tell. Moreover, since the model seems to have been solved in general, what is the point of the superstatistical approximation? To my taste, the results of this approximation confused me in Fig. 2.

My third general concern is that the narrative is rather technical and I'm afraid that I did not fully see the forest through the trees. I think I could get to a general understanding by studying this manuscript line by line and deriving everything, but an intuitive and user-friendly discussion is not given.

Fourth, in the quantitative finance literature, the feature of a stochastically fluctuating diffusion coefficient is known as the Heston model. A very readable paper that solves the Heston model is:

"Probability distribution of returns in the Heston model with stochastic volatility", A A Dragulescu and V M Yakovenko, Quantitative Finance 2, 443 (2002).

This work should be cited and the connection between the solution in this 2002 paper and the present manuscript should be pointed out. I realize that the 2002 paper focused on the probability distribution in an infinite domain, whereas the current manuscript focuses on first-passage properties in a finite domain.

On the positive side, I think that a suitably modified version of Fig. 2 could be extremely useful in encapsulating the results of the manuscript and in highlighting the difference between homogeneous diffusion and fluctuating diffusion.

Some questions about this key figure:

What's the point of showing the superstatistics results? They served to confused me.

Where is the vertical dashed line on panels 23 and 33 (matrix notation)? (I assume that it is beyond the right edge of the plot.)

The upper right of the figure is supposed to correspond to more homogeneous diffusion. However, there is still a large difference between the blue (heterogeneous) and black curves (homogeneous). In what limit to the homogeneous and heterogeneous curves approach each other? Can one show this approach in a quantitative and unambiguous way?

Obviously, the homogeneous and heterogeneous curves are generally different, as shown in the various panels. However, there should be a better way to illustrate the difference between the two sets of curves.

I also don't know what lesson I'm supposed to learn from the vertical dashed line.

To summarize, this paper is of high quality, but the presentation is overly technical, so that the main physical results are hard to extract. There are various lower-level shortcomings, as outlined in this report. If the authors can make the presentation more focused and user friendly, it is possible that the manuscript could be suitable for publication in Nature Communications.

Reviewer #2 (Remarks to the Author):

I am in favor of publishing this paper as it is. My comments about the results are summarized in a separate file.

Report on the manuscript "Diffusion-controlled reactions in dynamic heterogeneous media" by Yann Lanoiselée et al.

This paper discusses a timely issue of the concept of "diffusive-diffusivity" and its possible applications to model chemical kinetics under spatiotemporal disorder of the medium. Such a new class of diffusive dynamics has been recently detected experimentally in soft matter and biological systems and reported in a series of publications - also very well addressed and exemplified in the manuscript. An intriguing observation from those experiments is that probability density distribution derived from the trajectories of tracing particles has typically a Laplace form and converges to a Gaussian at times greater than the correlation time of fluctuations.

The desired model which is characterized by exponential tail at large displacements and converging slowly to the Gaussian distribution has been introduced and thoroughly analyzed by Lanoiselée and Grebenkov in a recent publication [50] where statistical and asymptotic properties of it have been derived. In the present manuscript Authors focus on spectral decomposition of the effective propagator from which - after integrating over the arrival points - the survival probability of a tracer inside given domain can be obtained and the PDF of first passage times deduced.

Principal results are reported in the SI section and are elegantly displayed in Fig.2 which presents the first passage time distributions from the center to a reactive (absorbing) boundary of a ball of radius R . The homogenization of the diffusion is there analyzed in terms of two dimensionless parameters characterizing disorder scale and strength. A clear deviation from the superstatistical version of the model is documented when analyzing broadening of the PDF and uprising of its short and long-time tails by dynamic disorder.

The paper is very interesting and can serve a broad community of researchers. The results are of importance to many experimental groups working on chemical kinetics in disordered, dynamic media. They are well discussed and properly linked to existing literature. I am in favor of publishing this manuscript in Nature Communications without any further amendements.

Reviewer #3 (Remarks to the Author):

The manuscript by Lanoiselee et al. discusses first passage times (FPT) in a dynamically heterogeneous medium as modeled by a generalization of the diffusing diffusivity approach. The new component here is a mathematical framework to compute the full FPT distribution under such a dynamics for domains of arbitrary shapes. The method relies on the observation that temporal and spatial dependencies of the corresponding Fokker-Planck equation can be separated within this model. The problem is then solved by employing a spectral decomposition of the Laplacian. The obtained scheme is used to compute FPT distributions from the center of a ball as function of disorder length scale and strength.

Title and abstract of the manuscript, in my view, are promising too much: "We investigate the effect of a spatio-temporal disorder onto diffusion-controlled reactions ..." The discussion of diffusion-controlled reactions is reduced to the analysis of first-passage times. Without doubt, the FPT and hitting distributions are essential ingredients to calculate reaction rates, and these quantities need to be reconsidered in the presence of crowding. Their specific ramifications, however, on the overall reaction kinetics have not been worked out. Thinking of motifs from gene expression, some crucial aspects are missing: rebinding effects, the limited availability of some species due to small copy numbers, and (anti-)cooperativity effects even in the simplest relevant reaction schemes. Going beyond FPT statistics and working out the full reaction kinetics (for a specific reaction) would indeed be a leap forward on the problem of reaction kinetics in the presence of anomalous diffusion.

The manuscript is clearly written and well structured, it presents nice work which is certainly of interest to others in the field of FPT statistics. However, I do not see how it would change thinking in the field. Once the idea to use a separation of variables is conceived, the formalism is rather straightforward in the context of partial differential equations. While the general treatment seems to be new, the same idea was used in a recently accepted manuscript by one of the authors; there, the eigenfunctions of the Laplacian corresponding to the cylindrical geometry were used:

Grebenkov, Metzler and Oshanin: Towards a full quantitative description of single-molecule reaction kinetics in biological cells, Phys. Chem. Chem. Phys., 2018, 10.1039/C8CP02043D

In the present manuscript, the given standard example of a ball falls even

behind that with respect to complexity. Instead, the authors should demonstrate that their new approach to FPT distributions is capable of solving a critical open problem, e.g., for a specific biochemical reaction scheme (see above).

A technical remark: the connection between the diffusing diffusivity model and the CIR process, eq. (1), is interesting (which is the result of ref. [50]).

While the positivity of the fluctuating $D(t) \geq 0$ is obvious in the former model, it is not so for the CIR process. In particular, positivity cannot be

guaranteed for the interesting regime $\nu < 1$. This aspect should be commented

on in the manuscript.

Similarly, a no-flux condition at $D=0$ is imposed in the supplement after eq. (S1). How is this justified from the SDE, eq. (1)?

Minor remarks:

Eq. (4): the time derivative should be indicated explicitly, the prime is not self-explaining. It should also be added that absorbing boundary conditions are assumed in eq. (4) to yield the survival probability.

The discussion of the flux [paragraph before eq. (5)] is rather vague and should be made clearer.

After eq. (6): "the decay rate is decreased by the factor $\dots \leq 1$ ". The rate is multiplied by a factor less than one, so it is decreased by the inverse of that factor.

REPLIES TO THE FIRST REVIEWER

First of all, we thank the reviewer for his/her very positive evaluation of our work and numerous remarks that helped us to further improve the manuscript. In the following, we provide very detailed replies to all raised points and discuss the related changes in the manuscript (also highlighted by red color). We hope that the revised version is now suitable for publication in Nature Communications.

My first concern is the overlap of the present manuscript with Ref. [50]. It appears that many of the technical details in the manuscript and the supplementary material were first derived in Ref. [50]. The authors need to clearly state what is new beyond Ref. [50] in the present manuscript.

The overlap with Ref. [50] is as minor as it can be. In Ref. [50], we proposed the Feller process (also known as CIR process in financial literature) as a particular model of diffusing diffusivity and investigated the properties of the resulting propagator in the free space. That work was *exclusively* focused on the diffusive transport and its various features (non-Gaussian character, ergodicity, etc.). In absence of any boundary or catalytic surface, no chemical reaction problem could be even formulated there. In the present manuscript, we introduce a reactive surface and thus couple the heterogeneous diffusive transport to the reaction kinetics. This is a conceptual step forward as compared to Ref. [50]. In the manuscript, only two elements are “borrowed” from Ref. [50]: (i) the diffusing diffusivity model and (ii) exact and asymptotic forms of the propagator in the free space and the method of its derivation. This mathematical ground is then used to derive most of the new results presented in the manuscript. In particular, we introduce the concept of first-passage times in the framework of dynamic heterogeneous diffusion and derive the short-time and long-time asymptotic behavior of the probability density of reaction times; we also study the impact of dynamic disorder on the macroscopic reaction rates, and other related questions. Finally, we stress that the general spectral decomposition in Eq. (2) is valid for an arbitrary diffusing diffusivity model and thus is independent of Ref. [50], except for the explicit form of the function $Y(t;\lambda)$ describing the dynamic disorder.

To highlight the distinctions between Ref. [50] and the manuscript, we significantly re-organized the discussion on pages 1-2. In particular, Ref. [50] (now Ref. [43]) is introduced in the same paragraph as the former works (now Refs. [38-42]) on diffusing diffusivity in the free space. In the second paragraph on page 2, we write explicitly that the former works did not involve any chemical kinetics. In the third paragraph (beginning of Section “Results”), we state explicitly that our manuscript couples heterogeneous diffusion with chemical kinetics on reactive targets. We expect that the above explanations and the re-organization remove any doubts concerning overlaps with Ref. [50].

A second issue is the non self-contained nature of the manuscript. In the text, there is reference to subordination and to superstatistics, both of which are treated in the SI. However, the meaning of these two concepts in the context of the new manuscript results is not given, as far as I can tell. Moreover, since the model seems to have been solved in general, what is the point of the superstatistical approximation? To my taste, the results of this approximation confused me in Fig. 2.

In the original version, most technical derivations based on subordination and superstatistics had been moved to the SI in order to make the manuscript itself shorter and easier for a broad readership. We agree with the reviewer that the meaning of these concepts was lacking in the main text.

Although the model has been solved exactly, the use of superstatistics remains very helpful for the analysis of the short-time behavior. Even for normal Brownian motion, the short-time behavior is always a tricky part of the analysis, given that one needs to account for infinitely many terms in the spectral decomposition. For instance, the approximation of the quantity of interest (e.g., the propagator) in a confined domain by the free Gaussian propagator turns out to be very useful. In the same vein, the superstatistical approach, which is getting more and more accurate as time goes to zero, greatly facilitates the analysis. The excellent quality of the obtained asymptotic results is confirmed by comparison with the exact solution.

As the detailed presentation of the superstatistics could make the manuscript too technical and long, we follow the reviewer's remark and remove the superstatistical approximation from Fig. 2 and from the main text. In turn, a new figure S1 is added to the SI to illustrate the accuracy of this approximation.

In contrast, we think that the subordination concept is fundamental and critically relevant to the manuscript. For this reason, we keep and extend the discussion of subordination in the main text. To make the text more coherent, this discussion is moved to the second part of the manuscript (into the "Discussion" section). We present the basic ideas of subordination and illustrate them by Figure 3 which has been moved from the SI to the main text. The subordination allows for various extensions of the present work to other models of dynamic disorder, as discussed in the text.

Finally, to make the manuscript more self-contained, the formula (10) for the probability density function of the first-passage times for a ball is moved from the SI to the main text. The explicit form of Eq. (10) also serves us to illustrate how our general approach can be applied to a particular case of interest.

My third general concern is that the narrative is rather technical and I'm afraid that I did not fully see the forest through the trees. I think I could get to a general understanding by studying this manuscript line by line and deriving everything, but an intuitive and user-friendly discussion is not given.

Following the reviewer's comment, we have paid particular attention to make discussions more intuitive and user-friendly. Throughout the text, we added many qualitative explanations and hints for readers. For instance, we explain more explicitly why the parameter $1/\nu$ represents the disorder strength (p.2); we give the meaning of the probability density function after Eq. (5); we explain the importance of considering this function for a small number of particles after Eq. (6) (with a new Ref. [57]); we discuss the meaning of short- and long-time asymptotic relations (p. 3); we justify our choice of the ball for illustrating the results (p. 4); we extend the discussion of subordination (p.5); we discuss limitations and perspectives of this work (p.5); we speak about the overall contribution of theoretical modeling in this field (p.6). This list of clarifications is far from

being complete and does not include numerous minor textual changes that we introduced for this purpose.

Finally, we structured the text by adding Sections “Results” and “Discussion”, as well as several subsections in the “Results” section, to facilitate reading.

Fourth, in the quantitative finance literature, the feature of a stochastically fluctuating diffusion coefficient is known as the Heston model. A very readable paper that solves the Heston model is: "Probability distribution of returns in the Heston model with stochastic volatility", A A Dragulescu and V M Yakovenko, *Quantitative Finance* 2, 443 (2002).

This work should be cited and the connection between the solution in this 2002 paper and the present manuscript should be pointed out. I realize that the 2002 paper focused on the probability distribution in an infinite domain, whereas the current manuscript focuses on first-passage properties in a finite domain.

We fully agree with the reviewer’s remark. In fact, the paper by Dragulescu et Yakovenko has inspired us in the first place to propose the Feller process as a model of diffusing diffusivity in the biophysical context in Ref. [50]. As the reviewer mentioned, this paper deals with diffusion in the free space, without any boundary. As it could not be directly related to chemical reaction problems (as we explained above), we did not cite it in the original version. However, we agree that this important contribution deserves a citation and has been included into the revision. We also note that the financial motivation led Dragulescu et Yakovenko to consider an analog of the geometric (or exponential) Brownian motion as a model of stocks returns, so that the derivations in that paper are different even from Ref. [50].

The paper has been included (new Ref. [42]) and cited on page 1, along with other works on diffusing diffusivity.

On the positive side, I think that a suitably modified version of Fig. 2 could be extremely useful in encapsulating the results of the manuscript and in highlighting the difference between homogeneous diffusion and fluctuating diffusion.

We thank the reviewer for this high evaluation.

Some questions about this key figure:

What's the point of showing the superstatistics results? They served to confused me.

The point was to show the accuracy of the superstatistical approximation at short times. We agree that these results may deviate reader’s attention from the main point and removed these curves as suggested by the reviewer. A new figure S1 to validate the accuracy of the superstatistical approximation has been added to the SI.

Where is the vertical dashed line on panels 23 and 33 (matrix notation)? (I assume that it is beyond the right edge of the plot.)

Yes, these dashed lines are beyond the plot (on the right side, as could be expected from panel 13).

The upper right of the figure is supposed to correspond to more homogeneous diffusion. However, there is still a large difference between the blue (heterogeneous) and black curves (homogeneous). In what limit do the homogeneous and heterogeneous curves approach each other? Can one show this approach in a quantitative and unambiguous way?

We thank the reviewer for pointing out this issue. The homogeneous diffusion is clearly recovered when either the time scale τ vanishes (with fixed σ) or the amplitude σ vanishes (with fixed τ), or both. In turn, the limits $\mu \rightarrow 0$ (with fixed $1/\nu$) and $1/\nu \rightarrow 0$ (with fixed μ), related to Fig. 2, are actually double limits, e.g., the former limit corresponds to $\sigma \rightarrow \infty$ and $\tau \rightarrow 0$. The asymptotic behavior is thus not evident and requires some additional analysis. We admit that putting the sentence “towards homogeneous diffusion” was not clear and in fact ambiguous.

Following the reviewer’s request, we undertook this analysis and wrote a new section III of the SI, in which the limiting behavior of the probability density function is studied in detail. The related discussion has also been modified in the main text on page 4.

Obviously, the homogeneous and heterogeneous curves are generally different, as shown in the various panels. However, there should be a better way to illustrate the difference between the two sets of curves.

As mentioned above, one of the original purposes of Fig. 2 was to show the accuracy of the superstatistical approximation. As the superstatistical approximation is now removed, panels of Fig. 2 compare various curves for heterogeneous diffusion (as parameters $1/\nu$ and μ vary) to the unique curve for homogeneous diffusion. We agree that the original way of presentation is not optimal anymore. For this reason, we replaced Fig. 2 by a new figure, in which one of the parameters is fixed while the other changes “continuously”. We hope that this new way of illustration is much better and fulfills the reviewer’s requirement.

I also don't know what lesson I'm supposed to learn from the vertical dashed line.

The vertical dashed line was meant to show a time scale τ below which the superstatistical approximation is getting more accurate. As Fig. 2 was modified, the vertical dashed lines were removed.

In summary, Figure 2 and the related text have been modified as suggested.

To summarize, this paper is of high quality, but the presentation is overly technical, so that the main physical results are hard to extract. There are various lower-level shortcomings, as outlined in this report. If the authors can make the presentation more focused and user friendly, it is possible that the manuscript could be suitable for publication in Nature Communications.

We thank again the reviewer for his/her evaluation and suggestions. We believe that the raised shortcomings have been fully removed in the revised manuscript.

REPLY TO THE SECOND REVIEWER

We thank the reviewer for his/her very positive evaluation of our work and recommendation for publication in the submitted form. Following the comments of other reviewers, we have revised the manuscript (changes are highlighted by red color and explained in the replies to two other reviewers). We hope that the reviewer would appreciate this revised version as well.

REPLIES TO THE THIRD REVIEWER

First of all, we thank the reviewer for his/her very positive evaluation of our work and numerous remarks that helped us to further improve the manuscript. In the following, we provide very detailed replies to all raised points and discuss the related changes in the manuscript (also highlighted by red color). We hope that the revised version is now suitable for publication in Nature Communications.

Title and abstract of the manuscript, in my view, are promising too much:

"We investigate the effect of a spatio-temporal disorder onto diffusion-controlled reactions ..." The discussion of diffusion-controlled reactions is reduced to the analysis of first-passage times. Without doubt, the FPT and hitting distributions are essential ingredients to calculate reaction rates, and these quantities need to be reconsidered in the presence of crowding. Their specific ramifications, however, on the overall reaction kinetics have not been worked out. Thinking of motifs from gene expression, some crucial aspects are missing: rebinding effects, the limited availability of some species due to small copy numbers, and (anti-)cooperativity effects even in the simplest relevant reaction schemes. Going beyond FPT statistics and working out the full reaction kinetics (for a specific reaction) would indeed be a leap forward on the problem of reaction kinetics in the presence of anomalous diffusion.

We agree that the main emphasis of the manuscript was put onto the first-passage times and related quantities (such as the macroscopic reaction rate in Eq. (6)). As pointed out by the reviewer, this is an essential ingredient and the first step in building a comprehensive theory of chemical reactions in dynamic complex environments. Even this first step involved substantial mathematical analysis and rather long and dense content of the SI. The analysis of further complications such as rebinding effects, partial reactivity, etc. is certainly important but it would be more suitable for future publications. We emphasize that such extensions are not straightforward. For instance, if there are N particles searching for a target site, one often assumes that they are independent and then the results for a single particle are trivially extended to multiple particles via basic combinatorics. One can formally employ the same arguments here and get the survival probability or the probability density of the first reaction time for many particles. However, the validity of such an approximation is less clear in the case of dynamically heterogeneous media. In fact, when two independent particles become close to each other, they experience similar interactions with the environment, and thus are expected to have the same diffusivity. In other words, encounters of particles can induce subtle correlations in their diffusivities. As a consequence, the assumption of independent diffusing diffusivity processes for each particle becomes debatable. Following the

reviewer's comment, we added this discussion as a new subsection of the SI (the last subsection of Sec. IV), to highlight the related difficulties. However, the study of the impact of such local correlations in diffusivities is an open difficult mathematical problem that requires a proper investigation which is beyond the scope of this paper.

Yet another example is the implementation of a partial reactivity κ of the target (which is closely related to rebinding). For normal diffusion, the partial reactivity is naturally implemented by replacing the Dirichlet boundary condition (immediate reaction upon encounter) by the Robin boundary condition in which the diffusive flux density toward the target is proportional to the reaction flux density: $-D \partial_n c = \kappa c$, where ∂_n is the normal derivative oriented outwards the domain, and c is a concentration. Formally, one can still use all the spectral decompositions presented in the manuscript, with the Laplacian eigenfunctions accounting for the Robin boundary condition. However, the microscopic probabilistic interpretation of these results becomes questionable. In fact, once a particle approaches the target, it has a random diffusivity D_t , given by the Feller process. In other words, at each moment when the particle is near the boundary, the diffusive flux density and thus the conventional Robin boundary condition are different. Can one use the Robin boundary condition in this situation? What should one put as the diffusivity D ? What would be the appropriate form of Robin boundary condition for heterogeneous diffusion? In our opinion, the conventional form of the Robin boundary condition for the marginal propagator $P(x,t|x_0)$ (and related quantities) is not valid, and one needs to work with the full propagator $P(x,D,t|x_0,D_0)$. As the current solution (presented in Sec. I of the SI) relies on the Laplace transform with respect to D , the Robin boundary condition would take an unusual differential form. One can see that these interesting questions raise new mathematical problems that require a proper analysis, using both mathematical and numerical tools.

At the same time, we agree with the reviewer that these points are important. Following the reviewer's remark, we made many changes in the manuscript. First, we clearly stated that our analysis is focused on *perfectly* reactive targets and *perfectly* reflecting obstacles. Second, we added a new paragraph on page 5, in which the points raised by the reviewer are discussed. In particular, we discuss the new level of complexity of diffusion-reaction problems in dynamic heterogeneous environments and related challenging problems on mathematical side. Third, we extended the SI (Sec. IV) by including the new computation of the survival probability for multiple particles and discussing related difficulties. Fourth, we replaced in the title, abstract and throughout the text the general term "diffusion-controlled reaction" by "diffusion-limited reaction", which is restricted to perfect reactions upon the first encounter.

Finally, we mention that a large body of the modern theoretical research remains focused on perfect reactions (see, e.g., Nature 450, 77 (2007), Nature Chemistry 2, 472-477 (2010), Nature Chemistry 4, 568-573 (2012), Nature 534, 356-359 (2016), which all treat only first-passage events). For instance, most narrow escape problems are formulated with mixed Dirichlet-Neumann boundary conditions, for which our formalism is applicable. The inclusion of other mechanisms presents an important and interesting perspective for future research.

However, I do not see how it would change thinking in the field.

We show that the effect of dynamic heterogeneity greatly impacts the distribution of the first-passage time and thus the chemical kinetics. It is not just an appropriate renormalization of the diffusion coefficient that would account for these effects. Both short-time and long-time asymptotic behaviors change dramatically. In other words, we demonstrate that ignoring these effects can be strongly misleading. Moreover, we show that, rather counter-intuitively, the dynamic disorder increases both short- and long-time tails of the probability density of the first-passage time, meaning that the disorder can actually be beneficial for a faster reaction by a single molecule. This is particularly important for living cells because actin waves and microtubule movements are known to constantly disturb the cytoplasm and can thus speed up some intracellular reactions, as suggested by our analysis. We think that these new concepts can potentially change thinking in the field (or at least they will shatter old dogmas and stimulate new developments).

Once the idea to use a separation of variables is conceived, the formalism is rather straightforward in the context of partial differential equations.

We agree with the reviewer that both the separation of variables and the subordination concept, once clearly formulated and conceived, allow one to “translate” many results of homogeneous diffusion to heterogeneous one. This was precisely the purpose of our paper, i.e., to provide a clear mathematical formalism to solve an *a priori* difficult problem.

We stress that the model of diffusing diffusivity and related results on chemical reactions also raise new challenges in the context of partial differential equations. We already mentioned two points: the collective search by multiple particles (that become correlated via diffusivity) and the partial reactivity (the proper formulation of the Robin boundary condition). These mathematical problems remain open. In this light, we believe that our manuscript can stimulate new developments in the mathematical field of stochastic processes and related partial differential equations. In this respect, we thank again the reviewer for pushing us in this direction that allowed us to reformulate some statements in the manuscript and make it attractive to even broader scientific community.

While the general treatment seems to be new, the same idea was used in a recently accepted manuscript by one of the authors; there, the eigenfunctions of the Laplacian corresponding to the cylindrical geometry were used: Grebenkov, Metzler and Oshanin: Towards a full quantitative description of single-molecule reaction kinetics in biological cells, *Phys. Chem. Chem. Phys.*, 2018, 10.1039/C8CP02043D

We are not sure to fully understand the reviewer’s point. The use of eigenfunctions in the context of chemical reactions is very broad (and old): almost all classical solutions to diffusion problems in bounded domains are obtained in terms of eigenfunctions. However, we do not see the relation to the PCCP manuscript, in which an approximate scheme for obtaining the distribution of first passage times was developed. This approximate scheme consisted in replacing the mixed Dirichlet-Neumann boundary condition by an effective inhomogeneous Neumann condition. The diffusion coefficient was constant, the concepts of diffusing diffusivity, subordination and superstatistics were even not mentioned in the PCCP paper. Finally, the PCCP paper did not actually deal with eigenfunctions, although a sort of spectral decomposition was obtained at the end by the inverse Laplace transform. In this light, we do not see any overlap between the PCCP

paper and the current manuscript, except for the joint topic of studying the distribution of reaction times. Perhaps, we missed something in the reviewer's comment.

In the present manuscript, the given standard example of a ball falls even behind that with respect to complexity. Instead, the authors should demonstrate that their new approach to FPT distributions is capable of solving a critical open problem, e.g., for a specific biochemical reaction scheme (see above).

The standard example of a ball is the most common theoretical model of a geometric confinement. In addition to obvious simplifications of this shape allowing one to get explicit solutions for many problems, the ball is a common shape for many applications. We certainly agree that a ball does not represent all possible geometric settings and, in particular, geometric complexity can play an important role. Nevertheless, we believe that the choice of a ball for illustrating the new features of heterogeneous diffusion was rightful and beneficial for the manuscript. First, it is simple enough to get the presentation accessible to a broad readership of Nature Communications. Second, it allows for obtaining explicit formulas which do not require numerical simulations; in other words, there is no additional complexity related to numerics. Third, it facilitates the comparison of these new formulas to classical results known for homogeneous diffusion; if we considered more complex shapes, one would need to compute numerically the results for homogeneous diffusion as well, thus eliminating the possibility of a straightforward and thus more convincing comparison. Fourth, even homogeneous diffusion in complex shapes far from being fully understood; as a consequence, it would be difficult to disentangle the features coming from heterogeneous diffusion and the features coming from geometric complexity; for instance, if we considered a small spherical target surrounded by a larger concentric impenetrable sphere, for which the problem remains exactly solvable, the presence of two geometric length scales – the radii of two spheres – would add a new dimension to the space of parameters and render Figure 2 of the manuscript less clear; here, we do not claim that these effects are not interesting (they are), but any additional geometric complexity would blur the clear impact of the dynamic disorder as we present it now. We continue working in this direction to show the impact of geometric complexity in a future publication.

These explanations have been added to the manuscript on page 4.

A technical remark: the connection between the diffusing diffusivity model and the CIR process, eq. (1), is interesting (which is the result of ref. [50]). While the positivity of the fluctuating $D(t) \geq 0$ is obvious in the former model, it is not so for the CIR process. In particular, positivity cannot be guaranteed for the interesting regime $\nu < 1$. This aspect should be commented on in the manuscript. Similarly, a no-flux condition at $D=0$ is imposed in the supplement after eq. (S1). How is this justified from the SDE, eq. (1)?

The question of positivity of $D(t)$ for our model (i.e., the CIR process) has been thoroughly discussed in Ref. [50] (now Ref. [43]). In general, the presence of the term $(D(t))^{1/2}$ in front of the noise dW_t in Eq. (1) ensures that the solution $D(t)$ is nonnegative, as shown by Feller in his seminal paper [44]. However, as it was first discussed by Feller and pointed out by the reviewer, $D(t)$ can be zero in the regime $\nu < 1$, in which case the process would stop. In order to avoid such unphysical situations, we impose the no-flux condition at $D=0$, instead of the absorbing condition.

As discussed in Ref. [50] (and also partly in Phys. Rev. E, 91 012123 (2015), which is now added to the SI), the no-flux condition resolves some “paradoxical” features of the CIR process, including the accumulation of probability at the point $D=0$. We thought that this discussion was too technical to be included into the manuscript, given that it was already provided in Ref. [50]. However, following the reviewer’s comment, we have added a brief discussion to Sec. I of the SI to clarify these important points. In particular, we present explicitly the no flux condition.

Minor remarks:

Eq. (4): the time derivative should be indicated explicitly, the prime is not self-explaining.

To avoid ambiguity as suggested by the reviewer, we write explicitly the expression for Y' in the form of the time derivative and also provide the explicit form of Y' in Eq. (5). This form can be also seen as an explicit definition of this notation. We hope that this clarification resolves the raised issue.

It should also be added that absorbing boundary conditions are assumed in eq. (4) to yield the survival probability.

In general, the boundary condition does not need to be absorbing. For instance, one can consider the situation when a part of the boundary is reflecting (Neumann boundary condition) while the other part is absorbing (Dirichlet boundary condition), as in the case of narrow escape problems. Following the reviewer’s remark, we added this clarification after Eq. (2).

The discussion of the flux [paragraph before eq. (5)] is rather vague and should be made clearer.

The discussion was clarified. Moreover, we also modified Eq. (5) (now Eq. (6)) to keep an arbitrary initial concentration of particles.

After eq. (6): "the decay rate is decreased by the factor ... ≤ 1 ". The rate is multiplied by a factor less than one, so it is decreased by the inverse of that factor.

Corrected.

We thank again the reviewer for his/her evaluation and suggestions. We hope that the raised shortcomings have been removed in the revised manuscript.

REVIEWERS' COMMENTS:

Reviewer #1 (Remarks to the Author):

I gave extensive comments on the first version of the manuscript and the authors have addressed most of them in a reasonable way. I now support the publication of this manuscript for Nature Communications.

Reviewer #2 (Remarks to the Author):

Authors made substantial effort by addressing criticism of reviewers and removing shortcomings raised in the evaluations. The response attached by them is detailed and covers all issues brought up in reviewers' comments. A new version of the manuscript seems by all means suitable for publication, especially given the timely subject of studies presented in.

Reviewer #3 (Remarks to the Author):

I thank the authors for their extensive and informative reply to my concerns.

The revised manuscript by Grebenkov et al. contains a series of substantial improvements over the original version. It is clear now that the manuscript focuses on the kinetics of independent reactions in the diffusion limit (perfectly reacting targets). Generalizations to imperfect targets and correlated reactions are outlined and concrete steps and open questions towards this difficult problem are identified. The authors have highlighted the conceptual innovations of their work, which indeed go well beyond the added ref. [57]. The consequences of the dynamically heterogeneous medium on the first passage time statistics have been exposed much clearer, which condenses in the greatly improved Figs. 2 and S3, which are now accompanied by a discussion based on the relevant length scales, thereby conveying physical insight.

In summary, all my concerns have been addressed convincingly and the present manuscript marks a significant step forward in the difficult general problem of reaction kinetics in crowded media. Without any further reservations, I recommend publication in Nature Communications.